# Generalists vs. Specialists: Evaluating LLMs on Highly-Constrained Biophysical Sequence Optimization Tasks

**Angelica Chen** [* 1 2]   **Samuel D. Stanton** [* 2]   **Frances Ding** [3]   **Robert G. Alberstein** [3]   **Andrew M. Watkins** [3]
**Richard Bonneau** [2]   **Vladimir Gligorijević** [2]   **Kyunghyun Cho** [2]   **Nathan C. Frey** [2]

## Abstract

Although large language models (LLMs) have shown promise in biomolecule optimization problems, they incur heavy computational costs and struggle to satisfy precise constraints. On the other hand, specialized solvers like LaMBO-2 offer efficiency and fine-grained control but require more domain expertise. Comparing these approaches is challenging due to expensive laboratory validation and inadequate synthetic benchmarks. We address this by introducing Ehrlich functions, a synthetic test suite that captures the geometric structure of biophysical sequence optimization problems. With prompting alone, off-the-shelf LLMs struggle to optimize Ehrlich functions. In response, we propose LLOME (Language Model Optimization with Margin Expectation), a bilevel optimization routine for online black-box optimization. When combined with a novel preference learning loss, we find LLOME can not only learn to solve some Ehrlich functions, but can even perform as well as or better than LaMBO-2 on moderately difficult Ehrlich variants. However, LLMs also exhibit some likelihood-reward miscalibration and struggle without explicit rewards. Our results indicate LLMs can occasionally provide significant benefits, but specialized solvers are still competitive and incur less overhead.

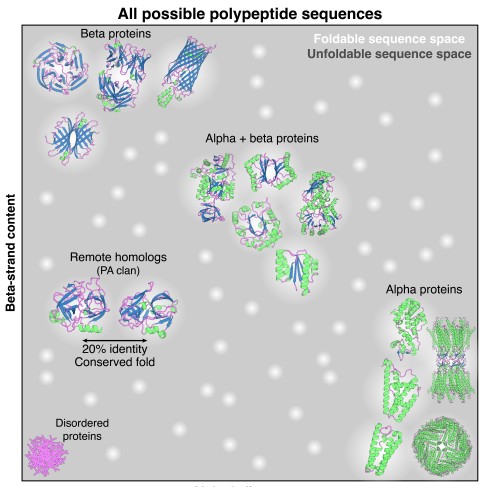

Figure 1: The space of all possible polypeptide sequences is vast, but only a tiny fraction forms stable, folded (*i.e.* feasible) proteins. Different protein families (alpha, beta, and mixed alpha/beta) generally occupy distinct regions of sequence space, but disordered proteins and remote homologs with conserved structure and low sequence identity illustrate the complexity of the protein design landscape.

## 1. Introduction

Despite their remarkable abilities, large language models (LLMs) often fail at tasks with fine-grained constraints — recent work has shown that even state-of-the-art models struggle to reliably generate text with a fixed number of words or to incorporate specific keywords or constraints (Garbacea & Mei, 2022; Sun et al., 2023; Yuan et al., 2024; Chen et al., 2024c). This limitation becomes especially critical in black-box biomolecule optimization problems (Figure 1), where even minor violations of biophysical constraints like protein stability or solubility can render a solution impossible to synthesize, purify, and assay (Hie et al., 2024; Ismail et al., 2024). While specialized solvers like LaMBO-2 address these constraints through careful modeling choices and architectural design (Gruver et al., 2024), adapting such solvers to new domains requires significant domain expertise and engineering effort. Recent work suggests that the

Work done at Genentech [1]Center for Data Science, New York University, New York City, U.S.A. [2]Prescient Design, Genentech, New York City, U.S.A. [3]Prescient Design, Genentech, San Francisco, U.S.A.. Correspondence to: Angelica Chen <angelica.chen@nyu.edu>, Samuel Stanton <stanton.samuel@gene.com>.

*Proceedings of the 42$^{nd}$ International Conference on Machine Learning*, Vancouver, Canada. PMLR 267, 2025. Copyright 2025 by the author(s).

Figure 2: An overview of Large language model optimization with Margin Expectation (LLOME). LLOME alternates between two optimization loops: (1) the outer loop trains the LLM on oracle-labeled data, and (2) the inner loop generates candidate sequences through iterative refinement without oracle access. At each outer loop iteration, the highest-ranked candidates are evaluated by the oracle to generate training data for the next iteration. LLOME enables effective optimization while minimizing expensive oracle queries through its bi-level structure.

capabilities of LLMs may be attainable at a far lower cost than previously thought (Zhu et al., 2024; Guo et al., 2025), and a deeper question remains: can improved preference learning methods help these models combine human-like flexibility with precise constraint satisfaction?

A fundamental challenge obstructing the development of LLMs as black-box optimization (BBO) solvers is evaluation. Unlike typical machine learning (ML) benchmarks for supervised and unsupervised models, optimization algorithms cannot be evaluated with a static dataset unless the search space is small enough to be exhaustively enumerated and annotated with the test function. Existing synthetic test functions commonly used in BBO research (Molga & Smutnicki, 2005) have very well-documented structure and solutions, making train-test leakage into pretrained LLM weights almost certain. Real-world black-box objectives by definition do not have formally characterized structure or solutions and are usually expensive to query. For instance, biomolecule optimization tasks require wet lab experiments for verification and chatbot systems require online user feedback, which is unsuitable for rapid development.

It is clear that test functions that are both more accessible and more difficult are needed for early-stage research and validation. We propose *Ehrlich functions*[1] as an idealized model of real biomolecule BBO tasks like antibody affinity maturation, building on principles from structural biology and biomolecular engineering experience. Ehrlich functions have adjustable difficulty and are always provably solvable; easy instances can be solved quickly by a genetic algorithm and used for debugging, but the same algorithm fails to solve harder instances after consuming over 500M function evaluations. These results can be reproduced in minutes on a single GPU. Importantly, Ehrlich functions are procedurally generated and not yet compromised by train-test leakage to pretrained LLMs. State-of-the-art LLM chatbots struggle to solve Ehrlich functions to optimality by prompting alone,

even when the prompt reveals the entire test function. We show that LLMs nevertheless *can* be taught to solve some Ehrlich functions when used to drive a bilevel optimization loop with online feedback, and are particularly effective when paired with a novel preference learning loss. Finally, a non-trivial closed-form objective allows us to deeply study preference learning itself, leading to new insights. In summary, our findings and contributions are as follows:

1. **Novel Test Functions for BBO and Preference Learning:** Ehrlich functions are accessible to all researchers, difficult for state-of-the-art solvers, and well-motivated by real biomolecule optimization problems.

2. **LLOME (Large Language Model Optimization with Margin Expectation)** We propose a bilevel optimization algorithm that allows LLMs to explore and learn new capabilities from online feedback when prompting fails.

3. **Margin-Aligned Expectation (MargE) Loss:** Our experiments on Ehrlich test functions motivate us to propose MargE, a novel training objective that maintains the simplicity of supervised fine-tuning (SFT) and direct preference optimization (DPO), yet outperforms them.

4. **New Insights into Preference Learning:** We find that preference-tuned LLM likelihoods do not necessarily correlate with the true objective. Furthermore, iterative preference tuning with ground-truth rewards outperforms training on preference pairs alone, and DPO in particular suffers from mode collapse and over-optimization.

5. **Comparisons between LLMs and specialized solvers:** We compare LLOME to LAMBO-2, a solver purpose-built for constrained discrete BBO. We find that LLOME can outperform or perform as well as LAMBO-2 on medium difficulty test functions, and is comparable on easier and harder variants, indicating that specialized models remain competitive after accounting for compute cost.

---

[1]Named after Paul Ehrlich, an early pioneer of immunology.

## 2. Background

We focus on pre-trained autoregressive large language models $\pi_\theta(x)$ parameterized by $\theta$. $\pi_\theta$ defines a probability distribution over discrete tokens $x \in \mathcal{V}$ for vocabulary $\mathcal{V}$. We can also define the likelihood of sequences $\mathbf{x} \in \mathcal{V}^*$ as $\pi_\theta(\mathbf{x}) = \prod_{t=1}^{|\mathbf{x}|} \pi_\theta(x_t|x_{<t})$, where $\mathcal{V}^*$ is the set of all concatenations of tokens in $\mathcal{V}$.

**Supervised Finetuning (SFT)** After pre-training, LLMs are typically finetuned on some task-specific dataset $\mathcal{D} = \{(\mathbf{x}_i, \mathbf{y}_i)\}_{i=1}^n$ consisting of pairs of input $\mathbf{x}$ and target $\mathbf{y}$ sequences. During SFT, $\pi_\theta$ is trained to minimize the negative conditional log likelihood of examples from $\mathcal{D}$:

$$\mathcal{L}_{\text{SFT}}(\theta) := \underset{\mathbf{x},\mathbf{y}\sim\mathcal{D}}{\mathbb{E}} - \log \pi_\theta(\mathbf{y}|\mathbf{x})$$

**Preference Learning** In some settings, LLMs are further trained to align their output distributions to a *reward distribution*, typically encoded with a reward model $r(\mathbf{x}) : \mathcal{V}^* \to \mathbb{R}$. This is frequently accomplished with reinforcement learning, where the LLM is trained via a policy gradient method to maximize the expected rewards $\mathbb{E}_{\mathbf{x}\sim\mathcal{D},\mathbf{y}\sim\pi_\theta(\cdot|\mathbf{x})} r(\mathbf{x}, \mathbf{y})$. The reward model is trained from a dataset of human preferences consisting of triples $(\mathbf{x}, \mathbf{y}_w, \mathbf{y}_l)$, where $\mathbf{x}$ is a prompt obtained from some offline dataset $\mathcal{X}$ and $\mathbf{y}_w$ and $\mathbf{y}_l$ are sampled from the current policy. The initial model is referred to as the *reference policy* $\pi_{\text{Ref}}$ and $\mathbf{y}_w, \mathbf{y}_l$ are assigned such that $\mathbf{y}_w$ is more preferred by human raters than $\mathbf{y}_l$. More recently, practitioners have added a KL regularizer to prevent the LLM from quickly over-optimizing, yielding a family of learning algorithms known as *reinforcement learning from human feedback* (RLHF; Ziegler et al., 2019; Stiennon et al., 2020), sharing a common objective:

$$\mathcal{L}_{\text{RLHF}}(\theta) := \underset{\substack{\mathbf{x}\sim\mathcal{X} \\ \mathbf{y}\sim\pi_\theta(\cdot|\mathbf{x})}}{\mathbb{E}} - r(\mathbf{x}, \mathbf{y}) + \beta\mathbb{KL}(\pi_\theta \| \pi_{\text{Ref}})$$

RLHF is commonly trained using Proximal Policy Optimization (PPO; Schulman et al., 2017), which involves considerable engineering complexity due to the need to train and coordinate four models ($\pi_\theta$, $\pi_{\text{Ref}}$, a reward model, and a critic model). Furthermore, RLHF-PPO is particularly sensitive to hyperparameter values and prone to training instabilities (Zheng et al., 2023b). To address some of these issues, Rafailov et al. (2023) proposed an offline version of RLHF known as Direct Preference Optimization (DPO). DPO skips reward modeling and directly trains on the preference triples with contrastive objective $\mathcal{L}_{\text{DPO}}(\theta)$:

$$\underset{\mathbf{x},\mathbf{y}_w,\mathbf{y}_l\sim\mathcal{D}}{\mathbb{E}} - \log\sigma\left(\beta\log\frac{\pi_\theta(\mathbf{y}_w|\mathbf{x})}{\pi_{\text{Ref}}(\mathbf{y}_w|\mathbf{x})} - \beta\log\frac{\pi_\theta(\mathbf{y}_l|\mathbf{x})}{\pi_{\text{Ref}}(\mathbf{y}_l|\mathbf{x})}\right),$$

where $\sigma$ is the sigmoid function. DPO often produces models with similar generative quality as RLHF-PPO, but involves notable tradeoffs such as faster over-optimization

(Rafailov et al., 2024a) and a distribution mismatch between the training dataset and policy outputs (Chen et al., 2024b; Tang et al., 2024). Nevertheless, DPO has become one of the most prevalent algorithms for offline alignment of LLMs. We provide further background on past work related to LLMs for optimization in Sec. 3.

## 3. Related Work

Our work combines insights from multiple areas of research, including discrete sequence black-box optimization, controllable text generation, and LLMs for optimization and scientific discovery. See Appendix A.1 for further discussion on related work.

**Discrete Sequence Black-Box Optimization** Many algorithms for discrete sequence optimization take inspiration from *directed evolution* (Arnold, 1998), a combination of random mutagenesis (a means to generate variants of the current solution) and high throughput screening (discriminative selection pressure). Researchers have explored many types of variant generators, including genetic algorithms (Back, 1996; Sinai et al., 2020), reinforcement learning (Angermueller et al., 2020), denoising with explicit discriminative guidance (Stanton et al., 2022; Maus et al., 2022; Gruver et al., 2024), and denoising with implicit discriminative guidance (Tagasovska et al., 2024). While these algorithms are all very general *in principle*, in practice a substantial amount of effort is required to actually implement these algorithms for new tasks due to changes in the problem setup. Our work investigates whether LLMs can provide a more generalizable approach that extends readily to new problem domains while maintaining competitive performance.

**LLMs for Optimization and Scientific Discovery** Prior work on LLMs for optimization has largely followed two approaches. The first uses LLMs to translate natural language descriptions into formal mathematical representations that can be solved by traditional optimizers (Ramamonjison et al., 2022; Ahmed & Choudhury, 2024). The second approach leverages LLMs directly as optimizers, often by embedding them within evolutionary algorithms (Romera-Paredes et al., 2023; Chen et al., 2024a) or using them for prompt-based optimization (Yang et al., 2024). Most closely related to our work is Ma et al. (2024), which also employs LLMs in a bilevel optimization loop. However, while they assume access to gradients through differentiable simulations, our method operates in the more challenging setting where only black-box evaluations are available. We also provide novel insights into how LLMs can improve their optimization capabilities through specialized training objectives, even without access to ground-truth rewards during the inner optimization loop.

**Controllable Text Generation (CTG)** CTG represents

a specialized case of optimization where the objective is to generate sequences with specific attribute values rather than maximizing a general objective function. While LLM prompting can effectively control high-level attributes (Brown et al., 2020), precise control remains challenging (Carlsson et al., 2022). Two primary approaches have emerged: control codes prepended during training (Keskar et al., 2019; Padmakumar et al., 2023; Raffel et al., 2020; Madani et al., 2023) and inference-time guidance using auxiliary models (Dathathri et al., 2020; Liu et al., 2021; Deng & Raffel, 2023; Dekoninck et al., 2024). Our work bridges CTG and optimization by viewing sequence constraints as part of the optimization problem, demonstrating that LLMs can learn to generate sequences satisfying precise constraints through our bilevel optimization framework.

## 4. Method

The goal of any optimization procedure is to find a maximizer $x^* \in \arg\max_{x \in \mathcal{F}} f(x)$ of an objective function $f : \mathcal{X} \to \mathbb{R}$ over a feasible set $\mathcal{F} \subset \mathcal{X}$. In the *black-box* optimization (BBO) setting, we only have access to zero-order information about $f$. We can query $f$ at different inputs, but we get no other information (e.g., derivative information). With an infinite query budget, we could find $x^*$ by brute force. In practice, limited resources usually constrain both human and artificial intelligences to strategies that iteratively refine a current solution $x_i$. This local, iterative approach aligns with the decision-theoretic concept of *satisficing* —seeking a "good enough" improvement rather than an exhaustive global optimum (Simon, 1956; Wilson, 2024). Furthermore, in many real high-dimensional search spaces, such iterative local optimization is surprisingly effective (Wu et al., 2023). The success of gradient-based training for neural networks, which finds solutions through iterative local updates, itself underscores the power of local optimization strategies.

### 4.1. The LLOME Algorithm

To adapt LLMs for the aforementioned BBO setting, we propose LLOME, which employs a bi-level optimization strategy for iteratively refining an LLM policy $\pi_\theta(\mathbf{y}|\mathbf{x})$ to generate improved sequences. This process alternates between improving the policy based on current data (the outer loop) and using the updated policy to generate and select new, high-potential candidates for black-box evaluation (the inner loop). The bi-level process unfolds as follows:

1. **Policy Improvement (Outer Loop).** At each iteration $i$, given the cumulative dataset $\mathbb{D}^{(i)}$ (containing all previously evaluated sequences and their scores), the outer loop tunes the policy parameters $\theta_i = \arg\min_\theta \mathcal{L}_{\text{train}}(\theta; \mathbb{D}^{(i)}, \pi_{\text{ref}})$. In our primary ap-

---

**Algorithm 1** LLOME, an approach for bilevel optimization of highly constrained sequence optimization problems with LLMs. We use $n_0 = 10$, $j = 2000$, and $T = 10$.

---

**Input:** Scoring function $f$; pretrained LLM $\pi_{\theta_0}$ parameterized by initial weights $\theta_0$; initial seed sequence $\mathbf{x}_0 \in \mathcal{F}$; $j$ number of test function evaluations per round; $T$ number of LLOME rounds.

$S \leftarrow \{(\mathbf{x}_0, f(\mathbf{x}_0))\}$
$\mathcal{X}_0 \leftarrow \text{GENETICALGORITHM}(\mathbf{x}_0, n_0)$ ▷ Seed with $n_0$ rounds of evolution.
$S_0 \leftarrow \{(\mathbf{x}, f(\mathbf{x})) \mid \mathbf{x} \in \mathcal{X}_0\}$ ▷ Score the initial candidates.
$S \leftarrow S \cup S_0$
$i \leftarrow 0$
**while** $i < T$ ▷ *Outer Loop*
**do**
$\quad \mathcal{D}_i \leftarrow \text{DATASETFORMATTING}(S_i)$
$\quad \theta_{i+1} \leftarrow \text{TRAIN}(\theta_i, \mathcal{D}_i)$
$\quad \mathcal{X}_{i+1} \leftarrow \text{ITERATIVEREFINEMENT}(\pi_{\theta_{i+1}}, S_i)$ ▷ Inner Loop
$\quad \mathcal{X}_{i+1} \leftarrow \text{FILTER}(\mathcal{X}_{i+1}, j)$ ▷ Filter $\mathcal{X}_{i+1}$ down to $j$ samples.
$\quad S_{i+1} \leftarrow \{(\mathbf{x}, f(\mathbf{x})) \mid \mathbf{x} \in \mathcal{X}_{i+1}\}$ ▷ Oracle labeling.
$\quad S \leftarrow S \cup S_{i+1}$
$\quad i \leftarrow i + 1$
**end**
**Output:** $\arg\max_{(\mathbf{x}, f(\mathbf{x})) \in S} f(\mathbf{x})$

---

proach, $\mathcal{L}_{\text{train}}$ is the **MargE** loss (Sec. 4.2), however other preference-based or supervised training objectives can also be used.

2. **Policy Execution (Inner Loop).** Using the updated policy $\pi_{\theta_i}$ from the outer loop, the inner loop generates and selects a new batch of candidate sequences for evaluation. This phase operates *without* direct calls to the black-box oracle $f$.

   - Input prompts $X_{\text{seeds}}^{(i)}$ are selected from the top-scoring historical sequences in $\mathbb{D}^{(i)}$.
   - The policy $\pi_{\theta_i}$ generates new candidate output sequences $\{\mathbf{y}_1^{(i)}, \ldots, \mathbf{y}_j^{(i)}\}$ from these seeds via multiple sampling from $\pi_{\theta_i}$. Heuristics such as adjusting sampling temperature are used to manage diversity and avoid premature collapse.
   - A subset $Y^{(i)} \subseteq \{\mathbf{y}_1^{(i)}, \ldots, \mathbf{y}_j^{(i)}\}$ is chosen by ranking with the policy likelihood $\pi_{\theta_i}(\mathbf{y}|\mathbf{x})$.

3. **Data Collection:** The selected candidates $Y^{(i)}$ are evaluated by the oracle $f$ to yield a new batch of labeled data $\mathbb{D}_{\text{batch}}^{(i+1)} = \{(\mathbf{y}, f(\mathbf{y})) \mid \mathbf{y} \in Y^{(i)}\}$. This new data is added to the cumulative dataset: $\mathbb{D}^{(i+1)} = \mathbb{D}^{(i)} \cup \mathbb{D}_{\text{batch}}^{(i+1)}$. The process then returns to the outer loop (Step 1) with the augmented dataset.

Algorithm 1 provides a high-level outline of LLOME. We collect an initial data package $\mathbb{D}^{(0)}$ from the history of a pre-solver (in our case, $n_0$ iterations of a genetic algorithm, details in Appendix A.6.3). In real applications, this data package may be provided to the user from historical records.

A key technique employed during policy improvement is dataset matching, an augmentation strategy proposed by Tagasovska et al. (2024). For all $\mathbf{x}_k \in \mathbb{D}^{(i)}$, we search $\mathbb{D}^{(i)}$ for other sequences $\mathbf{y}'_m$ in its local neighborhood (up to a distance cutoff $\Delta_{\mathrm{match}}$) that represent known improvements (i.e., $f(\mathbf{y}'_m) > f(\mathbf{x}_k)$). See Appendix A.6 for detailed subroutine descriptions. The modularity of the policy training objective $\mathcal{L}_{\mathrm{train}}$ in the outer loop enables us to benchmark different LLM fine-tuning strategies within the LLOME framework. In our experiments, we compare LLOME-MARGE against variants such as LLOME-SFT and LLOME-DPO.

## 4.2. Theoretical Motivation for MargE

Having described the higher-level overview of LLOME, we now motivate the design of our LLM training objective, MargE, from first principles.

To emulate an iterative refinement strategy with an LLM, we pose the task as learning a policy $\pi_\theta(\mathbf{y}|\mathbf{x})$ that responds with a refined solution $\mathbf{y} \in \mathcal{X}$ given a current prompt solution $\mathbf{x} \in \mathcal{X}$, aiming to maximize some measure of local improvement $r : \mathcal{X} \times \mathcal{X} \to \mathbb{R}$. Given a finite dataset of triplets $\{(\mathbf{x}_i, \mathbf{y}_i, r_i)\}_{i=1}^n$ generated through such an iterative process, we seek to learn the optimal policy for selecting improvements. Ideally, we would want to select $\mathbf{y}$ to deterministically maximize $r(\mathbf{x}, \mathbf{y})$. However, acknowledging our bounded compute resources (i.e., fixed model architecture and training budget) and incomplete information (finite training data), we invoke the *principle of maximum entropy* (Jaynes, 1957) as our guide for choosing the form of the optimal *stochastic* policy $\pi^*$. This principle leads us to the Boltzmann distribution $\pi^*(\mathbf{y}|\mathbf{x}) \propto \exp(\beta \cdot r(\mathbf{x}, \mathbf{y}))$, where $\beta > 0$ is the *rationality parameter* controlling the explore-exploit tradeoff. The Boltzmann distribution is known to maximize policy entropy subject to the constraint of achieving a certain expected reward (Ortega & Braun, 2013; Jeon et al., 2020), and is mathematically equivalent to the Bradley-Terry preference model (Luce et al., 1959).

To ensure stable learning and ground our policy, we regularize $\pi_\theta$ towards a prior policy $\pi_{\mathrm{ref}}$ (e.g., a base pre-trained LLM). Our theoretical policy objective is an instance of the generalized variational Bayes (GVB) framework (Knoblauch et al., 2019). We call our specific instantiation the **F**orward-**Re**verse **KL** (FReKL) loss, which we define as follows:

$$\mathcal{L}_{\mathrm{FReKL}}(\theta) := D_{\mathrm{KL}}(\pi_\theta \parallel \pi^*) + \lambda D_{\mathrm{KL}}(\pi_{\mathrm{ref}} \parallel \pi_\theta). \quad (1)$$

This objective is designed to achieve several key desiderata. First, the data term $D_{\mathrm{KL}}(\pi_\theta \parallel \pi^*)$ pushes the learned policy $\pi_\theta$ towards the optimal target policy $\pi^*$. Second, this formulation satisfies the Strong Interpolation Criteria (SIC; Hu et al., 2024), ensuring a smooth tradeoff between $\pi^*$ and the reference policy $\pi_{\mathrm{ref}}$ as governed by $\lambda \in [0, +\infty)$ (proof

in Appendix A.3). For the regularization term, we employ a mass-covering reverse KL divergence, $\lambda D_{\mathrm{KL}}(\pi_{\mathrm{ref}} \parallel \pi_\theta)$. This choice encourages $\pi_\theta$ to retain broad coverage over regions where $\pi_{\mathrm{ref}}$ assigns probability, which helps avoid premature policy collapse by maintaining exploratory breadth. Furthermore, the reverse KL *does not* heavily penalize $\pi_\theta$ for placing mass on optimal responses in the tails of $\pi_{\mathrm{ref}}$ and is naturally suited for training with off-policy data, as its expectation is taken with respect to $\pi_{\mathrm{ref}}$. In contrast, objectives regularized by a forward KL (e.g., $D_{\mathrm{KL}}(\pi_\theta \parallel \pi_{\mathrm{ref}})$ as in Steinberg et al. (2024)) may also satisfy SIC; however, they are *prior* mode-seeking and better suited for on-policy sample estimation. In the next section, we discuss a computationally tractable form of Eq. (1) and our specific choice of improvement measure $r(\mathbf{x}, \mathbf{y})$.

## 4.3. Computing the FReKL Loss

Direct optimization of Eq. (1) is challenging due to the KL terms involving the intractable partition function of $\pi^*$. To arrive at a computationally tractable form suitable for training with off-policy data, we expand the KL divergence terms and apply importance sampling, using samples from the reference policy $\pi_{\mathrm{ref}}$, yielding the following expression for $\mathcal{L}_{\mathrm{FReKL}}(\theta)$:

$$\mathop{\mathbb{E}}_{\substack{\mathbf{x} \sim \mathbb{D}_\mathbf{x}, \\ y \sim \pi_{\mathrm{Ref}}(\cdot|\mathbf{x})}} \left[ \frac{\rho_\theta^{\mathbf{xy}} - \lambda}{|\mathbf{y}|} \log \pi_\theta(\mathbf{y}|\mathbf{x}) - \rho_\theta^{\mathbf{xy}} \cdot \beta r(\mathbf{x}, \mathbf{y}) \right], \quad (2)$$

where $\mathbb{D}_\mathbf{x}$ is the distribution of input prompts, $\rho_\theta^{\mathbf{xy}} := \pi_\theta(\mathbf{y}|\mathbf{x})/\pi_{\mathrm{Ref}}(\mathbf{y}|\mathbf{x})$ is the importance weight likelihood ratio, and $|\mathbf{y}|$ normalizes the policy log-likelihood by the length of the response. The full derivation, along with a discussion of its design principles in comparison to DPO and RLHF, can be found in Appendix A.4.

## 4.4. The Margin Reward Function

Thus far we have worked with the FReKL loss in general form for an arbitrary reward function $r(\mathbf{x}, \mathbf{y})$. For iterative refinement behavior, we propose a specific reward structure that directly quantifies the notion of improvement. We define the *margin reward* as:

$$r_{\mathrm{margin}}(\mathbf{x}, \mathbf{y}) := \begin{cases} f(\mathbf{y}) - f(\mathbf{x}) & \text{if } f(\mathbf{y}) > f(\mathbf{x}), \\ 0 & \text{otherwise,} \end{cases} \quad (3)$$

where $f(x\mathbf{v})$ is the objective value of the prompt and $f(\mathbf{y})$ is the value of the response. Typically if $\mathbf{x} \notin \mathcal{F}$ (the feasible set) $f(\mathbf{x})$ is defined to be $-\infty$, however we set $f(\mathbf{x}) = 0$ for all infeasible $\mathbf{x}$ to avoid infinite rewards.

The specific form of $r_{\mathrm{margin}}(\mathbf{x}, \mathbf{y})$, particularly the clipping at zero, is important. While simply using $f(\mathbf{y}) - f(\mathbf{x})$ as the reward might seem intuitive, it can lead to unintended

policy behavior due to the translation-invariance property of Boltzmann distributions. Specifically, an unclipped reward implies $\pi(\mathbf{y}|\mathbf{x}) = \pi(\mathbf{y}|\mathbf{x}')$ for all pairs $\mathbf{x}, \mathbf{x}' \in \mathcal{X}$, meaning the preference for any response $\mathbf{y}$ is independent of the prompt and depends only on $f(\mathbf{y})$ (see Appendix A.1 for details). Such prompt-independent responses would not satisfy our goal of local refinement. Clipping the reward at 0 when $f(\mathbf{y}) \leq f(\mathbf{x})$ resolves this issue and interestingly resembles a margin loss. Margin-based rewards are popular in the Bayesian Optimization literature; however, improvements are usually calculated against a fixed baseline, e.g. the best-observed value (Močkus, 1975; Jones et al., 1998).

To avoid ambiguity, when the FReKL loss (Eq. 2) is combined with the margin reward function (Eq. 3), we will call the resulting policy objective the **Marg**in-Aligned **E**xpectation (MargE) loss in the remainder of this work.

# 5. Evaluation

We now introduce *Ehrlich functions*, a class of closed-form test functions that capture the geometric structure of biophysical sequence optimization problems while enabling fast evaluation and reproducible benchmarking. These functions maintain key characteristics of real fitness landscapes, non-additive mutational effects, while avoiding the fidelity-latency tradeoffs inherent in simulation-based approaches (Kellogg et al., 2011; Hummer et al., 2023).

## 5.1. Defining an Ehrlich Function

We provide a complete description of Ehrlich functions in Appendix A.2. In brief, given a token vocabulary $\mathcal{V}$ and the set of sequences $\mathcal{V}^L$ formed of concatenations of $L$ tokens in $\mathcal{V}$, we first define $\mathcal{F} \subset \mathcal{V}^L$, the set of *feasible sequences*. $\mathcal{F}$ is defined as the support of a discrete Markov process (DMP), more details of which are given in Appendix A.2. We also define a set of $c$ *spaced motifs* that represent biophysical constraints in specific regions of a sequence. These motifs are expressed as pairs of vectors $\{(\mathbf{m}^{(1)}, \mathbf{s}^{(1)}), \cdots, (\mathbf{m}^{(c)}, \mathbf{s}^{(c)})\}$ where $\mathbf{m}^{(i)} \in \mathcal{V}^k$, $\mathbf{s}^{(i)} \in \mathbb{Z}_+^k$, and $k \leq L//c$. An Ehrlich function $f$ then describes the degree to which a sequence $\mathbf{x} \in \mathcal{V}^L$ is feasible and possesses all motifs, modulated by a response function $g$. For $q \in [1, k]$ bits of precision, $f$ is expressed as:

$$f(x) = \begin{cases} \prod_{i=1}^{c} g \circ h_q(\mathbf{x}, \mathbf{m}^{(i)}, \mathbf{s}^{(i)}) & \text{if } \mathbf{x} \in \mathcal{F}, \\ -\infty & \text{otherwise.} \end{cases} \quad (4)$$

The function $h_q$ defines the degree to which $\mathbf{x}$ fulfills a given constraint $(\mathbf{m}, \mathbf{s})$, and is defined as follows:

$$h_q(\mathbf{x}, \mathbf{m}, \mathbf{s}) = \frac{1}{q}\left(\max_{\ell < L} \sum_{j=1}^{k} \mathbb{1}\left[x_{\ell+s_j} = m_j\right] //\frac{k}{q}\right).$$

Setting $q = k$ corresponds to a dense signal which increments $h_q(\mathbf{x}, \mathbf{m}^{(i)}, \mathbf{s}^{(i)})$ whenever an additional element of any motif has been fulfilled. We provide additional details in Appendix A.2 about how to ensure that all motifs are jointly satisfiable and that there exists at least one feasible solution that attains the optimal value of 1.0. We also provide further evidence in Appendix A.2 that $f$ is difficult to optimize with a GA, even with small $L$, $k$, and $c$ values.

## 5.2. Experimental Setup

We evaluate LLOME, LAMBO-2, and the GA on four Ehrlich functions with varying parameters and difficulties. To avoid potential confusion between different test functions, we propose the following naming convention: **Ehr(**$|\mathcal{V}|$, $L$**)-**$c$**-**$k$**-**$q$. The test functions we consider are as follows:

- $f_1$: **Ehr(8, 128)-8-8-8** (easy)
- $f_2$: **Ehr(32, 32)-4-4-4** (medium)
- $f_3$: **Ehr(32, 128)-4-4-4** (medium)
- $f_4$: **Ehr(32, 128)-8-8-8** (difficult)

We tune hyperparameters for only a single test function ($f_2$) and carry the best configuration across to all functions (more details provided in A.7 and A.8). See the Appendix (Fig. 12) for results on a test function with $q < k$.

**LLOME Details** We compare the performance of three different variants of LLOME (LLOME-SFT, LLOME-MARGE, LLOME-DPO) against that of a GA. The details of the genetic algorithm are given in Appendix A.6.3 and the training details for SFT, DPO, and MargE are given in Appendix A.7. Each LLOME variant is seeded with data from 10 rounds of the GA (*i.e.*, $n_0 = 10$). For LLOME-MARGE and LLOME-DPO, one round of SFT is trained before continuing with MargE and DPO training in future iterations. All three variants use the pre-trained model PYTHIA 2.8B (Biderman et al., 2023) as the base model. During the TRAIN step of each iteration of LLOME (step 1 of Alg. 1), we train the current checkpoint for one epoch. Lastly, we use $j = 2000$ test function evaluations per iteration of LLOME.

**LAMBO-2 Details** We also compare the performance of LLOME against LAMBO-2. LaMBO-2 is a black box optimization algorithm tailored to protein sequence design with wet lab validation in real antibody lead optimization settings (Gruver et al., 2024). We instantiate LaMBO-2 to jointly train an encoder shared between a generative discrete diffusion head and discriminative heads, which guide generation via their predictions of the reward of a sequence and whether it satisfies the problem constraints. We use a CNN architecture and train the model from random initialization (for hyperparameter details see Appendix A.8). In the experiments presented below, the LaMBO-2 model has a total of 314K parameters. To compare directly with LLOME, we seed LaMBO-2 with the same 10 rounds of GA designs and

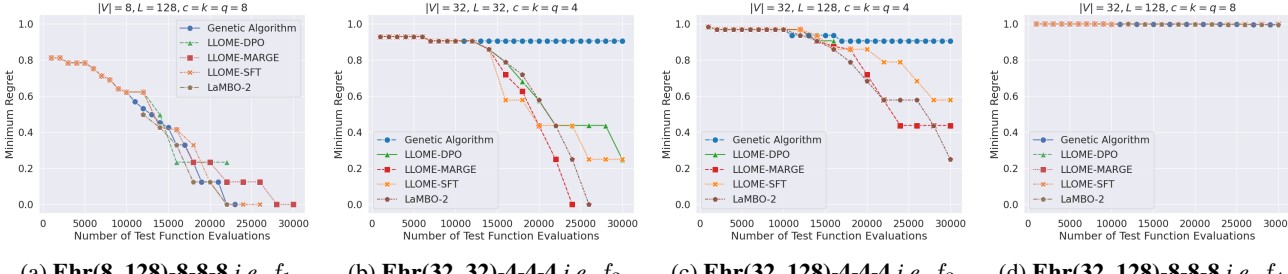

(a) **Ehr(8, 128)-8-8-8** *i.e.* $f_1$  (b) **Ehr(32, 32)-4-4-4** *i.e.* $f_2$  (c) **Ehr(32, 128)-4-4-4** *i.e.* $f_3$  (d) **Ehr(32, 128)-8-8-8** *i.e.* $f_4$

Figure 3: **On medium-difficulty tasks, LLOME-MARGE and LAMBO-2 outperform other methods. On very easy or difficult tasks, all methods perform comparably.** We show minimum simple regret achieved as a function of the number of test function evaluations for Ehrlich functions $f_1$, $f_2$, $f_3$, and $f_4$. Some lines end early due to early convergence to an optimum or generator collapse. The first 10K test function evaluations of every method correspond to the solutions produced by the GA pre-solver.

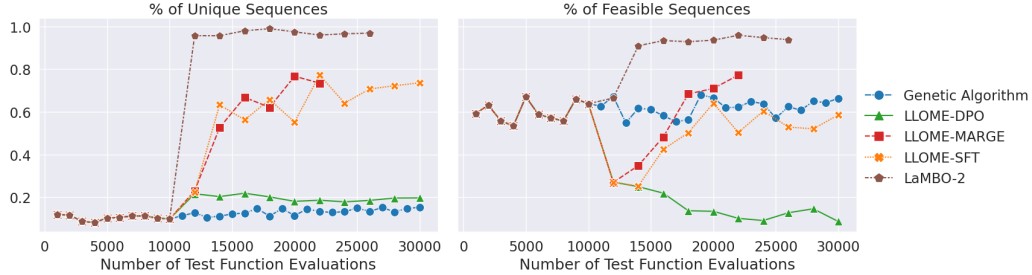

Figure 4: **LAMBO-2 produces the most diverse and feasible solutions but LLOME-MARGE balances diversity with accuracy.** The percentage of generated sequences for $f_2$ that are unique or feasible. Although LAMBO-2 achieves the most diverse and feasible solutions, LLOME-MARGE balances diversity with regret (Fig. 3b). The lines for LLOME-MARGE and LAMBO-2 end early because both discover the optimal solution early.

use $j = 2000$ test function evaluations per iteration.

**Metrics** We assess performance on Ehrlich function benchmarks with the simple regret metric: $\text{regret}(x) = 1 - f(x)$. We primarily show results on minimum simple regret over all optimization iterates, i.e. $\min_i \text{regret}(x_i)$, and the average regret of sequences in a given round. When applicable, we also evaluate each method's reward (Eq. 3), *i.e.*, how much the model's output improves over the input.

## 6. Results

### 6.1. Solver Benchmark Results

**State-of-the-art LLMs struggle to solve Ehrlich functions even with full problem specification.** As an initial test, we prompted OpenAI's `o1` (accessed on Jan. 28, 2025) and Google's Gemini 2 Flash Thinking (named 'Gemini 2.0 Flash Thinking Experimental 01-21' in Google AI Studio) models with a description of $f_2$, including the full transition matrix, all constraints, and a few in-context examples. (See prompt in A.5.) With only 8 samples, `o1` and Gemini Flash achieve minimum regrets of 0.9375 and 0, respectively. However, most outputs were either infeasible or had regret close to 1, indicating that this is still a challenging

problem for the LLMs, even full problem specification. In the rest of our experiments, we provide models only with pairs of sequence $\mathbf{x}$ and score $f(\mathbf{x})$.

**LLOME-MARGE can perform better than or comparable to specialized models in designing sequences under oracle label budget constraints.** In Figure 3, we plot the minimum regret achieved by each method as a function of the number of test function evaluations, under four different Ehrlich test functions with varying difficulty. First, we find that test function difficulty is essential for highlighting statistically significant differences between methods. On easy ($f_1$) or difficult ($f_4$) tasks, all methods achieve comparable performance, either finding the optimal solution very quickly or making virtually no progress. In contrast, on the medium difficulty test functions $f_2$ and $f_3$, performance is more differentiated. On $f_2$ LLOME-MARGE performs the best, whereas LAMBO-2 performs the best on $f_3$. The other methods – LLOME-SFT, LLOME-DPO, LAMBO-2, and the baseline GA – lag behind. Among the LLOME variants, LLOME-SFT and LLOME-MARGE achieve significantly higher rewards than LLOME-DPO (Figs. 14a, 14b). Test function $f_2$ (Figure 3) is particularly interesting as a case study, since the different methods have well-separated

regret curves. We thus focus on $f_2$ for further analysis.

**How do specialized models compare with generalist models?** At first glance, Figure 3 seems to show that generalist models like LLMs can outperform specialized models like LAMBO-2. However, one factor hidden in these plots is the computational cost of different methods. For relatively easy optimization problems, since the performance of various methods is similar, using a specialized model with 0.01% of the parameters of an LLM may be more practical. In addition, LAMBO-2 offers precise control over the sequence design space (through specifying number of tokens to mutate and the desired maximum edit distance). This allows domain experts to tune the algorithm with appropriate hyperparameters. Nonetheless, LAMBO-2 is sensitive to hyperparameter choice and requires custom tuning for each test function, whereas LLOME performs well across multiple functions without custom tuning. In Figs. 4 and 13 we plot the diversity and the feasibility of each method's designs over the course of optimization. We see that LLOME-MARGE strikes a careful balance of exploration and exploitation without custom tuning. However, LAMBO-2 proposes more diverse and feasible designs than LLOME-MARGE, but requires custom tuning for each test function.

**SFT- and MargE-trained LLMs generate unique and feasible sequences, but DPO-trained LLMs struggle.** Successfully solving a highly-constrained optimization problem requires that the model be able to generate a diverse array of feasible sequences. Compared to the GA, Figures 4 and 13 show that LAMBO-2, LLOME-MARGE, and LLOME-SFT produce significantly higher proportions of unique sequences. However, these three methods also experience an initial dip in the percentage of feasible outputs before learning to satisfy their constraints. In contrast, LLOME-DPO does not improve either the diversity or the feasibility of its outputs even when provided with more oracle labels. In some of our experiments, the LLOME-DPO pipeline ends prematurely due to producing an insufficient number of unique sequences to seed the next round. Like other past works (Pal et al., 2024; Rafailov et al., 2024b; Feng et al., 2024; Pang et al., 2024), we observe that the likelihood assigned by the DPO-trained LLM to both $y_w$ and $y_l$ continuously declines throughout training, implying that probability mass is moved to sequences outside of the training distribution. Since the percentage of infeasible sequences generated by LLOME-DPO increases over multiple iterations, it is likely that DPO moves some probability mass to infeasible regions of the sample space. As such, DPO may be ill-suited for training solvers for constrained problems.

Although LLMs are capable of generating new feasible sequences, we also find that they suffer from the classic explore versus exploit tradeoff. When the LLM makes a larger edit to the input sequence, the output is less likely to be feasible (Fig. 15). For both LLOME-SFT and LLOME-MARGE, an edit larger than 0.3 Hamming distance away from the input is less than 20% likely to be feasible. Since the $\Delta x$ threshold we set for the PropEn dataset formatting algorithm (Alg. 2) is 0.25, this is unsurprising. The LLM is never been trained on edits of larger than 0.25 Hamming distance. Overall, LLOME-MARGE exhibits the best tradeoff of all methods, producing the largest proportion of unique and feasible sequences for the smallest budget of test function evaluations and for moderately sized edits of Hamming distance $\leq 0.3$ (Fig. 15).

### 6.2. Understanding/Ablating LLOME

Here we summarize various aspects of LLOME's performance, deferring sections on LLOME's test-time extrapolation abilities and sensitivity to changes in hyperparameters, evaluation budget, and presolver to App. A.9.

**LLMs are moderately effective at ranking their own outputs.** The iterative refinement process requires that the LLM rank and filter its own outputs (Alg. 4), but we have not yet considered how effective the LLM is at selecting the best candidates. When compared to oracle selection (*i.e.* using the ground-truth score to select candidates), we find that the likelihood method often selects high-scoring but not the *highest* scoring candidates (Fig. 18). To better understand this gap between likelihood and oracle selection, we plot the calibration curve of regret versus model likelihood in Figure 5a. A perfectly-calibrated model would show a steep linear trend. The LLOME-DPO calibration curve monotonically decreases with regret, but also fails to generalize, producing the fewest sequences with regret outside of the training distribution. In contrast, LLOME-MARGE and LLOME-SFT assign higher likelihoods to lower regret sequences, but also exhibit some degree of overconfidence. Indeed, Fig. 17 shows that the likelihood and reward of SFT- and MargE-generated sequences are *inversely* correlated for high likelihoods exceeding 0.7. We hypothesize that LLMs may become increasingly miscalibrated as their outputs extend beyond the training distribution.

**When is explicit reward information required during training?** Since we have observed that LLOME-SFT and LLOME-MARGE perform similarly up to a certain point (Fig. 3), we might hypothesize that incorporating explicit reward values into the training objective is only necessary once the LLM is closer to the optimum. Since MargE requires a larger memory footprint than SFT (due to the need to store and compute likelihoods with both $\pi_\theta$ and $\pi_{\text{Ref}}$) and collecting ground truth rewards may be expensive, training the LLM with further rounds of SFT before switching to MargE would be more efficient than relying mostly on MargE training. We test this hypothesis via a multi-stage

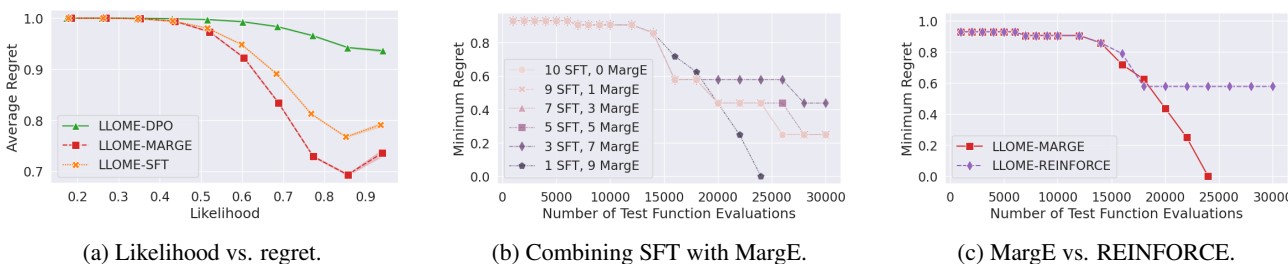

(a) Likelihood vs. regret.  (b) Combining SFT with MargE.  (c) MargE vs. REINFORCE.

Figure 5: We examine the design principles behind MargE, including calibration of likelihood vs. regret (5a), the importance of ground truth rewards (5b), and the difference between MARGE and policy gradient methods like REINFORCE (5c).

pipeline, where early rounds of LLOME use SFT training, and later rounds use MargE. We keep the total number of LLM training rounds constant at 10 (unless the LLM finds the optimal solution earlier), evaluate on test function $f_2$, and vary the proportion of rounds that use SFT versus MargE, as shown in Fig. 5b. We refer to the pipeline with $i$ rounds of SFT training followed by $j$ rounds of MargE training as LLOME-SFT$_i$-MARGE$_j$. We find that LLOME-SFT$_1$-MARGE$_9$ not only is the sole pipeline to find the optimal solution (Fig. 5b), but also achieves the best calibration curve (Fig. 19). In contrast, switching from SFT to MargE training at an intermediate point results in the worst performance. It appears that SFT and MargE may have conflicting loss landscapes – if we first train for a few rounds with the SFT loss, subsequently switching to MargE training impedes the LLM's progress.

**Does fulfilling the Strong Interpolation Criteria (SIC) matter for LLM training?** The drawbacks of SFT, DPO, and RLHF (as discussed in Sections 2 and A.1) motivated the design of MargE as a training objective that (1) uses the ground truth reward, (2) is less complex than RLHF, (3) reinforces the probability of high-reward outputs, (4) does not continuously increase output lengths, and (5) fulfills SIC (Hu et al., 2024). While criteria (1)-(4) have been discussed in prior sections, it remains to be seen whether (5) is indeed integral for training an LLM to optimize well.

To evaluate the impact of the SIC criteria on LLM training, we compare MARGE with REINFORCE (Williams, 1992), a policy gradient method that directly optimizes for maximal reward but does not fulfill SIC. We choose REINFORCE because it is similar in principle to MargE and has been shown to perform comparably (Ahmadian et al., 2024) to RLHF (Stiennon et al., 2020; Ziegler et al., 2019) and RL-free variants such as DPO (Rafailov et al., 2023) and RAFT (Dong et al., 2023). The effect of SIC is such that MargE smoothly interpolates between $\pi^*$ and $\pi_{\text{Ref}}$, whereas the KL-regularized REINFORCE algorithm interpolates between $\pi^\delta$ (a degenerate reward-maximizing distribution) and $\pi_{\text{Ref}}$, thereby violating SIC. In theory, moving the policy towards $\pi^\delta$ encourages collapse, which is undesirable in an online

optimization setting with finite data, where future rounds may require further exploration. We apply the same importance sampling and self-normalization from the MargE loss to the KL-regularized REINFORCE loss:

$$\mathbb{E}_{\substack{\mathbf{x}\sim\mathbb{D}_{\mathbf{x}},\\ \mathbf{y}\sim\pi_{\text{Ref}}(\cdot|\mathbf{x})}}\left[-\frac{\pi_\theta(\mathbf{y}|\mathbf{x})}{\pi_{\text{Ref}}(\mathbf{y}|\mathbf{x})}r(\mathbf{x},\mathbf{y})\log\pi_\theta(\mathbf{y}|\mathbf{x})-\lambda\frac{\log\pi_\theta(\mathbf{y}|\mathbf{x})}{|\mathbf{y}|}\right]$$

Fig. 5c shows that although LLOME-REINFORCE decreases the regret to some degree, it plateaus early and does not reach optimality, suggesting that the SIC criteria has meaningful influence on the LLM's optimization abilities.

## 7. Conclusions

Our work is a response to the lack of both non-trivial synthetic benchmarks for biophysical sequence optimizers and rigorous analyses of how LLMs perform on these highly-constrained optimization tasks in realistic settings. Our proposed test functions bear significant geometric similarities to real biophysical sequence optimization tasks and allow for rapid iteration cycles. In addition, although a wealth of work exists that adapts LLMs for biophysical tasks, few have studied the abilities of LLMs to adhere to hard constraints in realistic bi-level optimization settings. To that end, we propose and analyze LLOME, a framework for incorporating LLMs in bilevel optimization algorithms for highly constrained discrete sequence optimization problems. We show that in some settings, LLOME discovers lower-regret solutions than LAMBO-2 and a GA, even with very few test function evaluations. When combined with MargE training, LLOME is significantly more sample efficient than LLOME-SFT or LLOME-DPO, demonstrating its potential to be useful in data-sparse lab settings. However, in very easy or difficult tasks, specialized models have the advantage – they offer comparable regret for greater steerability and significantly lower computational cost. Our findings also highlight that LLMs can robustly extrapolate beyond their training data, but are occasionally miscalibrated and benefit from training with ground truth rewards.

## Impact Statement

This paper presents work whose goal is to advance the field of Machine Learning. There are many potential societal consequences of our work, none which we feel must be specifically highlighted here.

## Acknowledgements

We thank Natasa Tagasovska and the Prescient Design LLM team for valuable discussions and feedback during the development of this project. We additionally thank Miguel González-Duque for support with incorporating the LaMBO-2 method into the `poli-baselines` repository and for continued maintenance of `poli-baselines`. The authors also thank the Prescient Design Engineering team for providing HPC and consultation resources that contributed to the research results reported within this paper.

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

# A. Appendix

## A.1. Extended Related Work

**LLMs for Optimization and Scientific Discovery**   Much of the work on LLMs for optimization has been inspired by the design of classic black-box optimizers (BBO) such as genetic algorithms, bayesian optimizers, and random search methods. BBOs are characterized by a lack of information about the true objective function. Instead, only inputs and their corresponding objective values are provided to the optimizer, with no gradient or priors about the objective. Since these optimization problems are often expressible in formal mathematical, logical, or symbolic terms, many initial attempts at LLMs for optimization used the LLMs to first translate a natural language description of the problem into code or a modeling language, prior to passing this formalization into an auxiliary solver (Ramamonjison et al., 2022; Ahmed & Choudhury, 2024; Mittal et al., 2024; AhmadiTeshnizi et al., 2024). This approach was often quite effective, as it utilized the wealth of past work on LLMs for named entity recognition (Amin & Neumann, 2021; Ushio & Camacho-Collados, 2021; Wang et al., 2023b; Yan et al., 2019; Arkhipov et al., 2019, ; NER), semantic parsing (Drozdov et al., 2023; Gupta et al., 2018; Shaw et al., 2019; Shao et al., 2020; Rongali et al., 2020; Shi et al., 2021; Chen et al., 2022a; Stengel-Eskin et al., 2021), and code generation (Chen et al., 2021; Nijkamp et al., 2023b;a; Zhang et al., 2023b; Poesia et al., 2022). The NL4OPT competition (Ramamonjison et al., 2022), for example, decomposed LLMs-for-BBO into two subtasks: (1) the translation of a natural language description into a graph of entities and relations between the entities, and (2) a formalization of the optimization problem into a canonical logical reperesentation that could be solved by many commercial solvers. Winning approaches often employed ensemble learning (He et al., 2022; Wang et al., 2023a; Ning et al., 2023; Doan, 2022), adversarial learning (Wang et al., 2023a; Ning et al., 2023), and careful data pre-/post-processing and augmentation (He et al., 2022; Ning et al., 2023; Jang, 2022). Later work demonstrated that pre-trained LLMs like GPT-4 could achieve competitive performance on NLP4OPT without the NER stage (Ahmed & Choudhury, 2024), though the F1 score still trailed behind the state-of-the-art translation+NER approaches from the winning NL4OPT entries. Outside of NL4OPT, Gao et al. (2022), He-Yueya et al. (2023), AhmadiTeshnizi et al. (2024), and Mittal et al. (2024) use an LLM to formalize math, mixed integer linear programming, and combinatorial reasoning problems from a natural language description before offloading the solving to a Python interpreter or symbolic solver. Each work notes that the LLM performs better in this decomposed framework than through prompting alone.

Yet other approaches tackle LLMs-for-optimization directly with the LLM, without additional solvers. As Song et al. (2024) argues, LLMs offer both powerful in-context learning abilities and a flexible natural-language interface capable of expressing a wide variety of problems. Many techniques embed the LLM in an evolutionary algorithm, using the LLM as a mutation or crossover operator (Chen et al., 2024a; Guo et al., 2024b; Meyerson et al., 2024; Lehman et al., 2023; Nasir et al., 2024; Liu et al., 2024a; Lange et al., 2024; Liu et al., 2024b; Romera-Paredes et al., 2023). In this setting, the LLM provides a diversity of samples while the evolutionary algorithm guides the search towards high-fitness regions. This strategy is a form of bi-level optimization, in which two nested optimization problems (one nested within the other) are solved in alternation. It is common for the outer loop to optimize the model's parameters, and for the inner loop to optimize the model's outputs (Chen et al., 2022b; Guo et al., 2024c). Ma et al. (2024) combine this approach with feedback from physical simulations in the inner optimization loop to enable LLMs to complete various scientific optimization tasks, such as constitutive law discovery and designing molecules with specific quantum mechanical properties. Although their use of an LLM in a bi-level optimization loop is similar to ours, they directly train their parameters using differentiable simulations whereas we do not assume access to the gradients of the ground truth rewards. Our work also explores various aspects of LLM training that allow the LLM to improve its optimization abilities despite not having access to ground-truth rewards in the inner loop. Lastly, other approaches avoid gradient optimization altogether and focus purely on prompt-based optimization, demonstrating success on diverse tasks such as hill-climbing (Guo et al., 2024a), Newton's method (Nie et al., 2023), hyperparameter search (Zhang et al., 2023a), and prompt engineering (Khattab et al., 2024; Pryzant et al., 2023; Yang et al., 2024).

**Controllable Text Generation**   Controllable text generation (CTG) is a special case of optimization. Rather than searching for sequences that maximize the objective function, the goal is to produce a sequence with a particular attribute value (*e.g.* having a certain number of words, a more positive sentiment than the input, or a particular biological motif). In some aspects, this may be an easier task – the optimizer need not generate an optimal sequence that is likely outside the distribution of its training data. In others, this may also be more difficult. Targeting particular attribute values requires precise knowledge of the shape of the entire objective function, rather than only the neighborhood of the optima. Although LLM prompting is in itself considered a form of CTG (Radford et al., 2019; Brown et al., 2020), prompting alone tends to offer better control for

more open-ended, higher level instructions (*e.g.* "write a story in the style of Shakespeare") than for fine-grained constraints (*e.g.* "rewrite this sentence using only 10 words") (Carlsson et al., 2022).

One common approach is to use *control codes*, or unique strings pre-pended in front of a training example that indicates which attribute is represented in the example's target (Keskar et al., 2019; Padmakumar et al., 2023; Raffel et al., 2020; Madani et al., 2023). This approach is typically less generalizable to new attributes or instructions, due to the need to re-train the model. However, more recent work has shown that LLMs are capable of learning to use these control codes *in-context* (Zheng et al., 2023c; Zhou et al., 2023), simplifying the process by which new attributes can be introduced. A popular alternative technique is inference-time guidance, which uses auxiliary tools (Pascual et al., 2021) or models (Dathathri et al., 2020; Liu et al., 2021; Deng & Raffel, 2023; Dekoninck et al., 2024) to guide the LLM decoding process.

**Existing Benchmarks in Biophysical Domains** There are a few notable efforts to improve the state of sequence optimization benchmarks for biophysical domains.

**Small Molecules** In the small molecule domain, GuacaMol (Brown et al., 2019) and the Therapeutics Data Commons (TDC) (Huang et al., 2021) include simulation-based test functions for small molecule generation/optimization benchmarking. As we discussed in the main text, simulation-based test functions have significant barriers to entry ranging from computational resource requirements to software engineering concerns such as dependency management. If these simulations were in fact well-characterized, high-fidelity proxies for real molecule design objectives then these objections could be resolved, however at the time of writing it is difficult to determine 1) when a simulated task is "solved" and 2) what constraints are required to prevent ML methods from "hacking" the simulation and 3) to what degree simulation scores correspond at all to actual experimental feedback. Indeed, one could argue that if real molecule design objectives were sufficiently well-understood to characterize via simulation then the most effective approach to ML-augmented molecule design would be to simply approximate and accelerate those simulations rather than directly model experimental feedback.

**Large Molecules** In the large molecule domain, ProteinGym (Notin et al., 2023) assembles a collection of protein datasets and model baselines but is primarily targeted at evaluating offline generalization with a fixed dataset. The models from this benchmark could be used as "deep fitness landscapes" (i.e. an empirical function approximation optimization benchmark), with the corresponding limitations we discussed in the main text. Our work is most closely related to the FLEXS benchmarking suite (Sinai et al., 2020).[2] To our knowledge, FLEXS is the most comprehensive attempt to date to assemble a robust suite of benchmarks for large molecule sequence optimization, with benchmarks for DNA, RNA, and protein sequences from an array of combinatorially complete database lookups, empirical function approximators, and physics simulators. The Rough Mt. Fuji model (see below) is the only closed-form test function in the FLEXS suite. It is additive and hence trivial to model and solve. Hence our contribution can be seen as augmenting existing benchmark suites with test functions that are geometrically similar to real sequence optimization problems and also easy to install and cheap to evaluate.

**Models of Sequence Fitness in Theoretical Biology**

Geneticists have proposed theoretical models of biophysical sequence fitness and the geometry induced by random mutation and selection pressure, notably the mutational landscape model from Gillespie (2004), with more recent variants including the Rough Mt. Fuji model from Neidhart et al. (2014). These models are interesting objects of study, however those models assume mutational effects are either independent or additive, which disagrees with the correlated non-additive structure we observe empirically. These models also do not account for "fitness cliffs" (i.e. expression constraints that are highly sensitive to local perturbation and determine whether function is possible to observe experimentally). We implemented the Rough Mt. Fuji model as an additional test function and verified that a genetic algorithm can easily optimize it. Ehrlich functions can be seen as a constrained, non-additive mutational fitness landscape, and may be interesting objects for further theoretical analysis.

---

[2]https://github.com/samsinai/FLEXS

## A.2. Ehrlich Functions

### A.2.1. MOTIVATION FOR EHRLICH FUNCTIONS

Rigorous benchmarking is an essential element of good practice in science and engineering, allowing developers to evaluate new ideas rapidly in a low-stakes environment and thoroughly understand the strengths and weaknesses of their methods before applying them in costly, consequential settings.

While there has been a surge of investment into ML algorithms for applications like drug discovery, good benchmarks for those algorithms have proven elusive (Tripp et al., 2021; Stanton et al., 2022). Experimental feedback cycles in the life and physical sciences require expensive equipment, trained lab technicians, and can take months or even years. ML researchers require rapid feedback cycles, typically measured in minutes, necessitating proxy measures of success.

This need is particularly acute when evaluating black-box sequence optimization algorithms, which must produce a sequence of discrete states that optimizes a ground truth function. Unlike typical ML benchmarks for supervised and unsupervised models, optimization algorithms cannot be evaluated with a static dataset unless the search space is small enough to be exhaustively enumerated and annotated with the test function. Many researchers turn to simulation or empirical function approximation to provide test functions for larger, more realistic search spaces, however there is always a compromise between the highest possible fidelity (which may still be quite imperfect) and acceptable latency for rapid development.

We posit that well-designed synthetic (i.e. closed-form) test functions present many advantages as biomolecular design benchmarks, compared to simulations or empirical function approximations. First, we note that synthetic test functions have been universally used to test continuous optimization algorithms for decades (Molga & Smutnicki, 2005). Second, we note that optimization algorithms are designed to solve *any* test function belonging to a certain class. For example, gradient descent provably converges (under suitable assumptions) to the global optimum of any differentiable convex function. The key observation is that a test function need not correlate at all with downstream applications as long as there is shared structure (i.e. geometry). Thus the design principles we adopted to create Ehrlich functions were:

- Low costs/barriers to entry —a good benchmark should be inexpensive and easy to use.

- Well-characterized solutions —It should be easy to tell if a benchmark is "solved". Incremental progress towards better solutions should be reflected in the benchmark score.

- Non-trivial difficulty —a good benchmark should be challenging enough to motivate and validate algorithmic improvements. It should not be possible to solve with naïve baselines on a tiny resource budget.

- Similarity to real applications —while benchmarks inevitably require some simplification, a good benchmark should retain key characteristics of the desired application in a stylized, abstracted sense, otherwise the benchmark will not be relevant to the research community.

### A.2.2. LIMITATIONS OF EXISTING SEQUENCE OPTIMIZATION BENCHMARKS

With these criteria in mind, we categorize existing types of biophysical sequence optimization benchmarks and describe their limitations.

**Database Lookups** Database lookup test functions are constructed at substantial cost by exhaustively enumerating a search space and associating each element with a measurement of some objective, sometimes requiring large interdisciplinary teams of experimentalists and computationalists (Barrera et al., 2016b; Wu et al., 2016; Ogden et al., 2019; Mason et al., 2021; Chinery et al., 2024). Unfortunately this approach necessarily restricts the search space, and the correctness of the database itself cannot be completely verified without repeating the entire experiment.

**Empirical Function Approximation** Empirical function approximation benchmarks are related to database lookups since they start from an (incomplete) database of inputs and corresponding measurements. This type of test function returns an estimate from a statistical model trained to approximate the function that produced the available data (e.g. hidden Markov model sequence likelihoods, protein structure models, or "deep fitness landscapes") (Sarkisyan et al., 2016; Rao et al., 2019; Angermueller et al., 2020; Wang et al., 2022; Verkuil et al., 2022; Xu et al., 2022; Notin et al., 2023; Hie et al., 2024). Unfortunately empirical approximation is only reliable locally around points in the underlying dataset, and it is difficult to characterize exactly over which region of the search space the estimates can be trusted. As a result, blindly optimizing

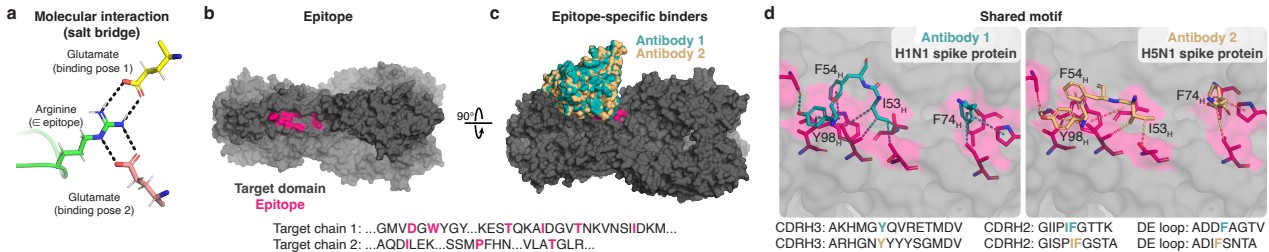

Figure 6: **(a)** Arginine and glutamate are complementary amino acids because they have a strong *salt bridge* interaction. **(b - c)** Antibodies that bind to a specific region of a target protein (the *epitope*) have many therapeutic and diagnostic uses. **(d)** Antibodies with different sequences can bind to the same epitope on two homologous proteins because they are structurally similar, which manifests as shared motifs in sequence space. Structures shown have RCSB codes 3gbn and 4fqi.

empirical function approximators often reveals an abundance of spurious optima that are easy to find but not reflective of the solutions we want for the actual problem (Tripp et al., 2021; Stanton et al., 2022; Gruver et al., 2024).

**Physics-Based Simulations** Simulations are a very popular style of benchmark, but current options all violate different requirements of a good benchmark. Most simulations are slow to evaluate, many are difficult to install, some require expert knowledge to run correctly, and yet still in the end simulations can admit trivial solutions that score well but are not actually desirable. For example, docking models have been proposed as test functions (Cieplinski et al., 2023), but they do not have well-characterized solutions and are easy to fit with deep networks (Graff et al., 2021). The primary appeal of simulations is a resemblance to real applications, however the resemblance can be superficial. $\Delta\Delta G$ simulations (Schymkowitz et al., 2005; Chaudhury et al., 2010) do not have a low barrier to entry, and yet the correlation of $\Delta\Delta G$ with real objectives (e.g., experimental binding affinity) is generally modest or unproven (Kellogg et al., 2011; Barlow et al., 2018; Hummer et al., 2023). Despite their difficulties, simulation benchmarks can be an important source of validation for mature methods for which we can justify the effort. However the limitations of simulations makes them especially unsuited for rapid method development, leading us to explore other alternatives.

**Closed-Form Test Functions** Closed-form functions have many appealing characteristics, including low cost, arbitrarily large search spaces, and amenability to analysis, however existing test functions for sequence optimization are so easy to solve that they are mostly used to catch major bugs. Simply put, designing a functioning protein is much, much harder than maximizing the count of beta sheet motifs (just one of many types of locally folded secondary structure elements in proteins) (Gligorijević et al., 2021; Gruver et al., 2024). The beta sheet test function highlights the main difficulty of defining closed-form benchmarks, namely not oversimplifying the problem to the point the benchmark becomes too detached from real problems.

A.2.3. EHRLICH FUNCTION DEFINITION AND CONNECTION TO REAL BIOPHYSICAL SEQUENCE DESIGN

Here we introduce Ehrlich functions and explain which specific aspects of real biophysical sequence design problems are captured by this function class, using antibody affinity maturation as a running example.

**Uniform random draws in sequence space are unlikely to satisfy constraints.** One of the first challenges encountered in black-box biophysical sequence optimization is a constraint on which sequences can be successfully measured. For example, chemical assays first require the reagents to be synthesized, and protein assays require the reagents be expressed by some expression system such as mammalian ovary cells. Popular algorithms like Bayesian optimization often assume the search space can be queried uniformly at random to learn the general shape of the function. Sequences of uniformly random amino acids generally do not fold into a well-defined structure and cannot be purified, meaning that we cannot measure anything about the objective function (e.g. binding affinity). Unfortunately constraints like protein expression cannot currently be characterized as a closed-form constraint on the sequence. We simplify and abstract this feature of biophysical sequences with the notion of a feasible set of sequences $\mathcal{F}$ with non-zero probability under a discrete Markov process (DMP) with transition matrix $A \in \mathbb{R}_+^v \times \mathbb{R}_+^v$,

$$\mathcal{F} = \{\mathbf{x} \in \mathcal{X} \mid A[x_{\ell-1}, x_\ell] > 0 \; \forall \ell \geq 2\}, \tag{5}$$

where $\mathcal{X} = \mathcal{V}^L$ is the set of all sequences of length $L \geq 2$ that can be encoded with states $\mathcal{V}$, with $|\mathcal{V}| = v$.

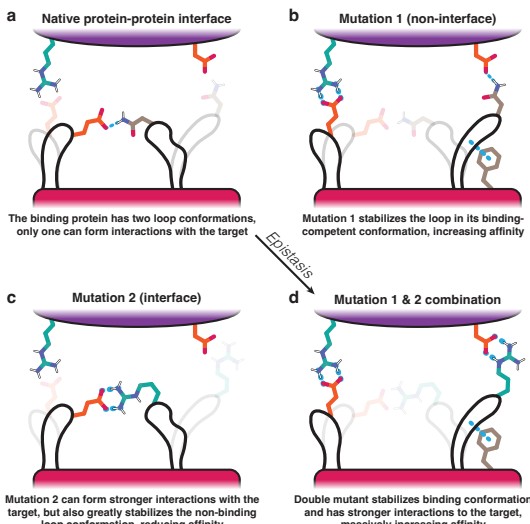

Figure 7: Illustration of an epistatic second-order interaction.

Note that if sequences are drawn uniformly at random, then assuming at least one entry of $A$ is zero (i.e. at least one state transition is infeasible), we have

$$\mathbb{P}[\mathbf{x} \notin \mathcal{F}] \geq \sum_{\ell=1}^{L//2} \left(1 - \frac{1}{v^2}\right)^{\ell} \frac{1}{v^2},$$

$$= 1 - \left(1 - \frac{1}{v^2}\right)^{L//2},$$

where $//$ denotes integer division. If we choose $L$ large enough we will see uniform random draws fall outside $\mathcal{F}$ with high probability. See Appendix A.2.4 for further details on our procedure to generate random ergodic transition matrices with infeasible transitions.

**Non-additive, position-dependent sensitivity to perturbation.** By construction, any sequence optimization problem can be written as minimizing the minimum edit distance to some set of optimal solutions $\mathcal{X}^*$. In the antibody engineering context $\mathcal{X}^*$ is not a singleton but a set of solutions that all satisfy a notion of *complementarity* with the target antigen of interest (more specifically the target *epitope*). As a simple example, if the epitope has an arginine residue, then placing a glutamate residue on the antibody creates the possibility of a *salt bridge* (see Fig. 6). Furthermore, the formation of a salt bridge in this example requires that we place the glutamate at specific *positions* on the antibody sequence that are in contact with the epitope (i.e. on the *paratope*). One of the reasons there are many possible solutions to the antibody-antigen binding problem is the absolute position of an amino acid in sequence space can vary as long as the resulting structure is more or less the same (i.e. there are two or more *structural homologs*). The functional effect of changes to the antibody sequence are not only non-additive, but can exhibit state-dependent higher-order interactions, a phenomenon known as *epistasis* (Fig. 7).

We abstract these features of biophysical sequence optimization by specifying the objective as the satisfaction of a collection of $c$ *spaced motifs* $\{(\mathbf{m}^{(1)}, \mathbf{s}^{(1)}), \ldots, (\mathbf{m}^{(c)}, \mathbf{s}^{(c)})\}$, where $\mathbf{m}^{(i)} \in \mathcal{V}^k$ and $\mathbf{s}^{(i)} \in \mathbb{Z}_+^k$ for some $k \leq L//c$. Given a sequence $\mathbf{x}$, we can represent the degree to which $\mathbf{x}$ satisfies a particular $(\mathbf{m}^{(i)}, \mathbf{s}^{(i)})$ with $q \in [1, k]$ bits of precision as follows:

$$h_q(\mathbf{x}, \mathbf{m}^{(i)}, \mathbf{s}^{(i)}) = \max_{\ell < L} \left( \sum_{j=1}^{k} \mathbb{1}\{x_{\ell+s_j^{(i)}} = m_j^{(i)}\} \right) // \left(\frac{k}{q}\right) / q. \tag{6}$$

The quantization parameter $q$ allows us to control the *sparsity* of the objective signal (note that $q$ must evenly divide $k$). Taking $q = k$ corresponds to a dense signal which increments whenever one additional element of the motif is satisfied. Taking $q = 1$ corresponds to a sparse signal that only increments when the whole motif is satisfied. For example, if $k = 2$

Figure 9: Here we show how the difficulty of the test problem can be controlled by varying Ehrlich function parameters, keeping the optimizer fixed to the robust GA baseline defined in Sec. A.6.3. Starting from a fixed set of reference parameters we vary each parameter individually. For this optimizer, the problem difficulty depends most strongly on the quantization parameter $q$. The x-axis is defined in *millions* (M) of Ehrlich function evaluations, demonstrating the difficulty of these Ehrlich functions, even for a small number of short motifs.

and $q = 2$ then Eq. equation 6 can assume the values 0, 0.5, or 1. If $k = 2$ and $q = 1$ then Eq. equation 6 can only assume the values 0 or 1.

The defining characteristic of epistasis is not merely non-additivity (which we can model via quantization and a product-of-motif-checks parameterization), but also *non-monotonicity* as a function of motif satisfaction. In simple terms, we simply mean that a beneficial mutation in one sequence context can be deleterious in another. We model non-monotonic mutational effects through the introduction of a *response function* $g : [0, 1] \to \mathbb{R}$. In particular we propose a cubic function $g(h) = ah^3 - ah^2 + h$, where $a \in \mathbb{R}_+$ is the *epistasis factor* (see Fig. 8). When $a = 0$, $g(h) = h$, corresponding to a linear response. When $a = 4$, $g(0) = g(0.5)$, which means satisfying half of the motif scores the same as satisfying none of it. While $a > 4$ can be used, one must be careful to guarantee that the optimal value still holds, i.e. $f^* = 1$, and $f$ will no longer be strictly non-negative. Note this reponse function is symmetric with respect to all motif elements, as there is no notion of some motif elements being more "robust" to epistasis than others. While this modeling decision does not entirely agree with empirical evidence, it greatly simplifies the implementation, and captures the key non-monotonic behavior we desire. In the experiments for

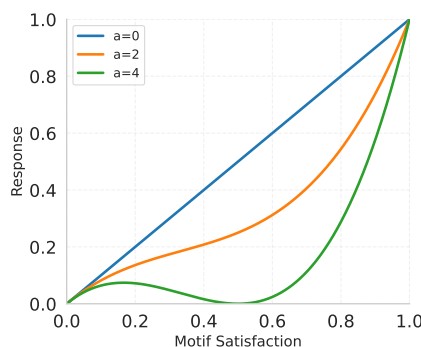

Figure 8: Motif satisfaction response function $g(h) = ah^3 - ah^2 + h$.

this paper we took $a = 0$ for all test functions, as we found the resulting optimization problems already sufficiently difficult to induce interesting differences in the optimizers, leaving an investigation of the effect of taking $a > 0$ for future work.

We are now ready to define an *Ehrlich function* $f : \mathcal{V}^L \to (-\infty, 1]$, which quantifies with precision $q$ the degree to which $\mathbf{x}$ *simultaneously* satifies all $(\mathbf{m}^{(i)}, \mathbf{s}^{(i)})$ if $\mathbf{x}$ is feasible, and is negative infinity otherwise.

$$f(\mathbf{x}) = \begin{cases} \prod_{i=1}^{c} g \circ h_q(\mathbf{x}, \mathbf{m}^{(i)}, \mathbf{s}^{(i)}) & \text{if } \mathbf{x} \in \mathcal{F} \\ -\infty & \text{else} \end{cases}. \tag{7}$$

Note that we must take some care to ensure that 1) the spaced motifs are *jointly satisfiable* (i.e. are not mutually exclusive) and 2) at least one feasible solution under the DMP constraint in Eq. equation 5 attains the global optimal value of 1. See Appendix A.2.5 for details.

One major advantage of procedurally generating specific instances of Ehrlich functions is we can generate as many distinct instances of this test problem as we like. In fact it creates the possibility of "train" functions for algorithm development and hyperparameter tuning and "test" functions for evaluation simply by varying the random seed. These functions can also be defined with arbitrary levels of difficulty, as shown in Fig. 9. However, defining a random instance that is nevertheless provably solvable takes some care in the problem setup, which we now explain.

A.2.4. CONSTRUCTING THE TRANSITION MATRIX

Here we describe an algorithm to procedurally generate random ergodic transition matrices $A$ with infeasible transitions. A finite Markov chain is ergodic if it is *aperiodic* and *irreducible* (since every irreducible finite Markov chain is positive recurrent). Irreducibility means every state can be reached with non-zero probability from every other state by some sequence of transitions with non-zero probability. We will ensure aperiodicity and irreducibility by requiring the zero entries of $A$ to have a banded structure. For intuition, consider the transition matrix

$$\begin{bmatrix} 0.4 & 0.3 & 0 & 0.3 \\ 0.3 & 0.4 & 0.3 & 0 \\ 0 & 0.3 & 0.4 & 0.3 \\ 0.3 & 0 & 0.3 & 0.4 \end{bmatrix}$$

Recalling that $v$ is the number of states, we can see that every state $x$ communicates with every other state $x'$ by the sequence $x \to (x+1) \mod v \to \cdots \to (x'-1) \mod v \to x'$. We also see that the chain is aperiodic since every state $x$ has a non-zero chance of being repeated.

To make things a little more interesting we will shuffle (i.e. permute) the rows of a banded structured matrix (with bands that wrap around), but ensure that the diagonal entries are still non-zero. Note that permuting the bands does not break irreducibility because valid paths between states can be found by applying the same permutation action on valid paths from the unpermuted matrix. We will also choose the non-zero values randomly, using the shuffled banded matrix only as a binary mask $B$ as follows:

$$\overset{\text{(banded matrix)}}{\begin{bmatrix} 1 & 1 & 0 & 1 \\ 1 & 1 & 1 & 0 \\ 0 & 1 & 1 & 1 \\ 1 & 0 & 1 & 1 \end{bmatrix}} \xrightarrow{\text{shuffle}} \begin{bmatrix} 1 & 0 & 1 & 1 \\ 1 & 1 & 1 & 0 \\ 1 & 1 & 0 & 1 \\ 0 & 1 & 1 & 1 \end{bmatrix},$$

$$\xrightarrow{\text{diag}=1} \begin{bmatrix} 1 & 0 & 1 & 1 \\ 1 & 1 & 1 & 0 \\ 1 & 1 & 1 & 1 \\ 0 & 1 & 1 & 1 \end{bmatrix}$$

$$= B.$$

Now we draw the transition matrix starting with a random matrix with IID random normal entries, softmaxing with temperature $\tau > 0$ to make the rows sum to 1, applying the mask $B$, and renormalizing the rows by dividing by the sum of the columns after masking.

$$Z = \overset{\text{(randn matrix)}}{\begin{bmatrix} +1.41 & +1.67 & -1.52 & +0.63 \\ -0.35 & +0.45 & +0.86 & -0.49 \\ +1.42 & -1.31 & -0.31 & +1.43 \\ -0.02 & +1.55 & -0.26 & +1.13 \end{bmatrix}},$$

$$\xrightarrow{\text{softmax}} \begin{bmatrix} 0.36 & 0.46 & 0.02 & 0.16 \\ 0.13 & 0.30 & 0.45 & 0.12 \\ 0.44 & 0.03 & 0.08 & 0.45 \\ 0.10 & 0.49 & 0.08 & 0.33 \end{bmatrix},$$

$$\xrightarrow{\odot B} \begin{bmatrix} 0.36 & 0 & 0.02 & 0.16 \\ 0.13 & 0.30 & 0.45 & 0 \\ 0.44 & 0.03 & 0.08 & 0.45 \\ 0 & 0.49 & 0.08 & 0.33 \end{bmatrix},$$

$$\xrightarrow{\text{norm}} \begin{bmatrix} 0.66 & 0 & 0.04 & 0.30 \\ 0.15 & 0.34 & 0.51 & 0 \\ 0.44 & 0.03 & 0.08 & 0.45 \\ 0 & 0.55 & 0.09 & 0.36 \end{bmatrix} = A.$$

We can also verify that $A$ is ergodic numerically by checking the Perron-Frobenius condition,

$$(A^m)_{ij} > 0, \ \forall i, j, \tag{8}$$

where $m = (v - 1)^2 + 1$, $A^1 = A$, and $A^b = A A^{b-1}$ for all $b > 1$. In our example, if $v = 4$ then $m = 10$ and we verify on a computer that

$$A^{10} = \begin{bmatrix} 0.33 & 0.23 & 0.17 & 0.27 \\ 0.33 & 0.23 & 0.17 & 0.27 \\ 0.33 & 0.23 & 0.17 & 0.27 \\ 0.33 & 0.23 & 0.17 & 0.27 \end{bmatrix}$$

### A.2.5. CONSTRUCTING JOINTLY SATISFIABLE SPACED MOTIFS

Here we describe how to procedurally generate $c$ spaced motifs of length $k$ such that the existence of a optimal solution $\mathbf{x}^*$ with length $L$ with non-zero probability under a transition matrix $A$ generated by the procedure in Appendix A.2.4 can be verified by construction. If we simply sampled motifs completely at random from $\mathcal{V}^k$ we cannot be sure that a solution attaining a global optimal value of 1 is actually feasible under the DMP constraint.

First we require that $L \geq c \times k$. Next to define the motifs, we draw a single sequence of length $c \times k$ from the DMP defined by $A$ (the first element can be chosen arbitrarily). Then we chunk the sequence into $c$ segments of length $k$, which defines the motif elements $\mathbf{m}^{(i)}$. This ensures that any motif elements immediately next to each other are feasible, and ensures that one motif can transition to the next if they are placed side by side.

Next we draw random offset vectors $\mathbf{s}^{(i)}$. The intuition here is we want to ensure that an optimal solution can be constructed by placing the spaced motifs end-to-end. If we fix $c \times k$ positions to satisfy the motifs, there are $L - c \times k$ "slack" positions that we evenly distribute (in expectation) between the spaces between the elements of each motif. We set the first element of every spacing vector $s_1^{(i)}$ to 0, then set the remaining elements to the partial sums of a random draw from a discrete simplex as follows:

$$\mathbf{w}^{(i)} \sim \mathcal{U}\left( \left\{ \mathbf{w} \in \mathbb{R}^{k-1} \mid \sum w_i = 1 \right\} \right). \tag{9}$$

$$s_{j+1}^{(i)} = s_j^{(i)} + 1 + \lfloor w_j^{(i)} \times (L - c \times k) \rfloor // c. \tag{10}$$

Finally, recall that in Appendix A.2.4 we ensured that self-transitions $x \rightarrow x$ always have non-zero probability. This fact allows us to construct a feasible solution that attains the optimal value by filling in the spaces in the motifs with the previous motif elements.

As a concrete example, suppose $L = 8$, $c = 2$, and $k = 2$ (hence $s_{\max} = 3$) and we draw the following set of spaced motifs:

$$\begin{bmatrix} 0 & 3 & 1 & 2 \end{bmatrix} \rightarrow \begin{bmatrix} 0 & 3 \\ 1 & 2 \end{bmatrix} = \begin{bmatrix} \mathbf{m}^{(1)} \\ \mathbf{m}^{(2)} \end{bmatrix}, \tag{11}$$

$$\begin{bmatrix} \mathbf{s}^{(1)} \\ \mathbf{s}^{(2)} \end{bmatrix} = \begin{bmatrix} 0 & 3 \\ 0 & 3 \end{bmatrix}. \tag{12}$$

$$\tag{13}$$

An optimal solution can then be constructed as follows:

$$\mathbf{x}^* = \begin{bmatrix} 0 & 0 & 0 & 3 & 1 & 1 & 1 & 2 \end{bmatrix}$$

Note that this solution is only used to *verify* that the problem can be solved. In practice solutions found by optimizers like a genetic algorithm will look different. Additionally if $L \gg c \times k$ then the spaced motifs can often be feasibly interleaved without clashes.

### A.2.6. DEFINING THE INITIAL SOLUTION

Optimizer performance is generally quite sensitive to the choice of initial solution. In our experiments we fixed the initial solution to a single sequence of length $L$ drawn from the DMP.

A.2.7. EHRLICH BENCHMARK RANK CORRELATION WITH REAL-WORLD DATA BENCHMARKS

To validate the real-world applicability of Ehrlich functions as an optimizer benchmark, we compared the performance of 64 variants of our genetic algorithm on Ehrlich functions versus three well-established lookup-based biological test functions: TFBind8 (Barrera et al., 2016a), TrpB (Papkou et al., 2023), and DhfR (Johnston et al., 2024). For each benchmark, we evaluate the median cumulative regret (estimated from 8 trials) of 64 variants of our GA with different hyperparameter settings. Each hyperparameter variant $(p_m, p_r, \alpha)$ was an element of the Cartesian product $\{0.0625, 0.125, 0.25, 0.5\} \times \{0.0625, 0.125, 0.25, 0.5\} \times \{0.0625, 0.125, 0.25, 0.5\}$. We then computed rank correlations between algorithm performance on these biological benchmarks vs. comparable Ehrlich functions. As shown in Table 1, the strong rank correlations are evidence supporting the claim that Ehrlich functions effectively capture the structure of real biological sequence optimization problems.

|  | DhfR | TFBind8 | TrpB |
|---|---|---|---|
| **Ehr(4, 4)-2-2-2** | 0.75 | 0.75 | 0.61 |
| **Ehr(20, 8)-2-2-2** | 0.86 | 0.89 | 0.73 |

Table 1: Spearman correlation coefficients of median regret achieved by our genetic algorithm variants on Ehrlich functions versus lookup-based biological test functions.

## A.3. Proof of Strong Interpolation Criteria for the Policy Objective

The policy objective is defined as:

$$\mathcal{L}(\theta) := D_{\mathrm{KL}}(\pi_\theta \parallel \pi^*) + \lambda D_{\mathrm{KL}}(\pi_{\mathrm{ref}} \parallel \pi_\theta), \tag{14}$$

where $\pi_\theta$ is the learned policy, $\pi^*$ is the optimal target policy (Boltzmann distribution based on rewards), $\pi_{\mathrm{ref}}$ is the reference prior policy, and $\lambda \geq 0$ is the interpolation hyperparameter. We assume that $\pi_\theta$, $\pi^*$, and $\pi_{\mathrm{ref}}$ are probability distributions over the same action space $\mathcal{Y}$ (conditioned on a state/prompt $x$, which we omit for brevity in the distributions). We also assume that the family of policies parameterized by $\theta$ is sufficiently flexible to represent both $\pi^*$ and $\pi_{\mathrm{ref}}$.

The Strong Interpolation Criteria (SIC) requires:

1. $\lim_{\lambda \to 0^+} \arg\min_{\pi_\theta} \mathcal{L}(\theta) = \pi^*$

2. $\lim_{\lambda \to \infty} \arg\min_{\pi_\theta} \mathcal{L}(\theta) = \pi_{\mathrm{ref}}$

3. For $\lambda \in (0, \infty)$, the optimal $\pi_\theta$ (denoted $\pi_\theta^{(\lambda)}$) interpolates between $\pi^*$ and $\pi_{\mathrm{ref}}$.

### A.3.1. PROOF OF SIC CONDITION 1 ($\lambda \to 0^+$)

As $\lambda \to 0^+$, the objective function in Eq. (14) becomes:

$$\lim_{\lambda \to 0^+} \mathcal{L}(\theta) = \lim_{\lambda \to 0^+} (D_{\mathrm{KL}}(\pi_\theta \parallel \pi^*) + \lambda D_{\mathrm{KL}}(\pi_{\mathrm{ref}} \parallel \pi_\theta))$$
$$= D_{\mathrm{KL}}(\pi_\theta \parallel \pi^*) + 0 \cdot D_{\mathrm{KL}}(\pi_{\mathrm{ref}} \parallel \pi_\theta)$$
$$= D_{\mathrm{KL}}(\pi_\theta \parallel \pi^*).$$

The Kullback-Leibler divergence $D_{\mathrm{KL}}(\pi_\theta \parallel \pi^*)$ is non-negative, i.e., $D_{\mathrm{KL}}(\pi_\theta \parallel \pi^*) \geq 0$. It achieves its minimum value of 0 if and only if $\pi_\theta(y) = \pi^*(y)$ for almost all $y \in \mathcal{Y}$. Therefore,

$$\lim_{\lambda \to 0^+} \arg\min_{\pi_\theta} \mathcal{L}(\theta) = \pi^*.$$

This satisfies the first condition of SIC.

### A.3.2. PROOF OF SIC CONDITION 2 ($\lambda \to \infty$)

As $\lambda \to \infty$, we analyze the objective function $\mathcal{L}(\theta) = D_{\mathrm{KL}}(\pi_\theta \parallel \pi^*) + \lambda D_{\mathrm{KL}}(\pi_{\mathrm{ref}} \parallel \pi_\theta)$.

Consider the case where $\pi_\theta = \pi_{\mathrm{ref}}$. In this scenario, $D_{\mathrm{KL}}(\pi_{\mathrm{ref}} \parallel \pi_\theta) = D_{\mathrm{KL}}(\pi_{\mathrm{ref}} \parallel \pi_{\mathrm{ref}}) = 0$. The objective function evaluates to:

$$\mathcal{L}(\theta)|_{\pi_\theta = \pi_{\mathrm{ref}}} = D_{\mathrm{KL}}(\pi_{\mathrm{ref}} \parallel \pi^*) + \lambda \cdot 0 = D_{\mathrm{KL}}(\pi_{\mathrm{ref}} \parallel \pi^*).$$

This value is a finite constant (assuming $\pi_{\mathrm{ref}}$ and $\pi^*$ have overlapping support such that the KL divergence is well-defined and finite).

Now, consider any policy $\pi_\theta \neq \pi_{\mathrm{ref}}$ such that $D_{\mathrm{KL}}(\pi_{\mathrm{ref}} \parallel \pi_\theta) > 0$. As $\lambda \to \infty$, the term $\lambda D_{\mathrm{KL}}(\pi_{\mathrm{ref}} \parallel \pi_\theta)$ will tend to $\infty$, because $D_{\mathrm{KL}}(\pi_{\mathrm{ref}} \parallel \pi_\theta)$ is a positive constant for this $\pi_\theta$. Thus, for $\pi_\theta \neq \pi_{\mathrm{ref}}$ (where $D_{\mathrm{KL}}(\pi_{\mathrm{ref}} \parallel \pi_\theta) > 0$):

$$\lim_{\lambda \to \infty} \mathcal{L}(\theta) = D_{\mathrm{KL}}(\pi_\theta \parallel \pi^*) + \lim_{\lambda \to \infty} \lambda D_{\mathrm{KL}}(\pi_{\mathrm{ref}} \parallel \pi_\theta) = \infty.$$

To minimize $\mathcal{L}(\theta)$ as $\lambda \to \infty$, the policy $\pi_\theta$ must be chosen such that the term growing with $\lambda$ is minimized, which means $D_{\mathrm{KL}}(\pi_{\mathrm{ref}} \parallel \pi_\theta)$ must be 0. This occurs if and only if $\pi_\theta(y) = \pi_{\mathrm{ref}}(y)$ for almost all $y \in \mathcal{Y}$. Therefore,

$$\lim_{\lambda \to \infty} \arg\min_{\pi_\theta} \mathcal{L}(\theta) = \pi_{\mathrm{ref}}.$$

This satisfies the second condition of SIC.

A.3.3. PROOF OF SIC CONDITION 3 (INTERPOLATION FOR $\lambda \in (0, \infty)$)

For any finite, positive $\lambda$, the objective function $\mathcal{L}(\pi_\theta) := D_{\mathrm{KL}}(\pi_\theta \,\|\, \pi^*) + \lambda D_{\mathrm{KL}}(\pi_{\mathrm{ref}} \,\|\, \pi_\theta)$ is minimized by a policy $\pi_\theta^{(\lambda)}$. We established that $\pi^*(y|x) \propto \exp(\beta \cdot r(x, y))$ is a Boltzmann distribution, and $\pi_{\mathrm{ref}}$ is the reference policy.

The objective $\mathcal{L}(\pi_\theta)$ is a sum of two non-negative divergence terms. The first term, $D_{\mathrm{KL}}(\pi_\theta \,\|\, \pi^*)$, pulls $\pi_\theta$ towards $\pi^*$. The second term, $\lambda D_{\mathrm{KL}}(\pi_{\mathrm{ref}} \,\|\, \pi_\theta)$, pulls $\pi_\theta$ towards $\pi_{\mathrm{ref}}$ (in a mass-covering sense).

If we operate directly in the space of probability distributions (assuming our model family $\Pi_\Theta = \{\pi_\theta\}$ is rich enough to contain the solutions), $L(\pi)$ is a strictly convex function of the distribution $\pi \in \Pi_\Theta$ for any $\lambda \geq 0$ (as both KL terms are convex in $\pi$, and the first is strictly convex if $\pi^*$ is fixed, and the second is convex in $\pi$). Thus, for any $\lambda \geq 0$, there exists a unique minimizing distribution $\pi^{(\lambda)}$ in a suitable convex space of distributions.

The first-order optimality condition for the unconstrained problem (in the space of distributions, subject to $\sum_y \pi(y) = 1$) for $\mathcal{L}(\pi)$ is found by setting the functional derivative with respect to $\pi(y)$ to zero:

$$\frac{\delta\mathcal{L}(\pi)}{\delta\pi(y)} = \left(\log\frac{\pi(y)}{\pi^*(y)} + 1\right) + \lambda\left(-\frac{\pi_{\mathrm{ref}}(y)}{\pi(y)}\right) + \mu = 0$$

where $\mu$ is the Lagrange multiplier for the normalization constraint. This implies that the optimal distribution $\pi^{(\lambda)}(y)$ satisfies:

$$\log\pi^{(\lambda)}(y) - \log\pi^*(y) - \lambda\frac{\pi_{\mathrm{ref}}(y)}{\pi^{(\lambda)}(y)} + (1 + \mu) = 0$$

$$\pi^{(\lambda)}(y) \propto \pi^*(y)\exp\left(\lambda\frac{\pi_{\mathrm{ref}}(y)}{\pi^{(\lambda)}(y)}\right). \quad (*)$$

This equation implicitly defines the unique optimal distribution $\pi^{(\lambda)}(y)$. The solution $\pi^{(\lambda)}$ to the implicit equation $(*)$ varies continuously with $\lambda$. This is because the objective function $\mathcal{L}(\pi)$ changes continuously with $\lambda$, and the minimizer of a strictly convex function typically depends continuously on such parameters. Thus, the optimal policy $\pi_\theta^{(\lambda)}$ (which is the $\pi_\theta$ in our model family that best approximates $\pi^{(\lambda)}$) smoothly interpolates from $\pi^*$ to $\pi_{\mathrm{ref}}$ as $\lambda$ varies from $0^+$ to $\infty$. This satisfies the third condition of SIC. $\qquad\square$

## A.4. Derivations

Although DPO has recently become a popular preference learning method due to its relative simplicity and competitive results, its offline contrastive objective suffers from a number of drawbacks. Firstly, since DPO optimizes for an off-policy objective, a DPO-trained model rapidly over-optimizes, resulting in generations that decline in quality despite continued improvements in offline metrics (Rafailov et al., 2024a; Chen et al., 2024b). DPO models also fail at ranking text according to human preferences (Chen et al., 2024b) and tend to decrease the probability mass assigned to preferred outputs (Pal et al., 2024; Rafailov et al., 2024b; Feng et al., 2024; Pang et al., 2024). As training continues, DPO generations also increase in length, even if the quality of the generations does not necessarily improve (Singhal et al., 2024). Additionally, when the reference model already performs well on a particular subset of the input domain, DPO cannot achieve the optimal policy without deteriorating performance on that subset (Hu et al., 2024). Lastly, DPO does not make use of absolute reward values – instead, it simply assumes that $r(x, y_w) > r(x, y_l)$ for all $(x, y_w, y_l)$ in the training dataset, but does not use information about *how much better* $y_w$ is than $y_l$.

RLHF, on the other hand, involves steep technical complexity and frequently exhibits training instabilities (Zheng et al., 2023a; Wang et al., 2024; Casper et al., 2023). Hu et al. (2024); Korbak et al. (2022b) additionally show that RLHF's objective interpolates between $\pi_{\mathrm{Ref}}$ and a degenerate distribution $\pi^\delta$ that places all probability mass on a single reward-maximizing sequence, thereby promoting generator collapse. Indeed, much past work (Kirk et al., 2024; Zhou et al., 2024; Padmakumar & He, 2024) has illustrated the low diversity of RLHF-generated text.

**Derivation of the MargE Objective**   To derive a training objective that fulfills SIC, we follow Hu et al. (2024) and propose an objective that takes the following general form:

$$\arg\min_\theta\left[\mathbb{KL}(\tilde{\pi}_\theta\|\pi^*) + \lambda\mathbb{KL}(\pi_{\mathrm{Ref}}\|\tilde{\pi}_\theta)\right] \tag{15}$$

where $\pi^*$ is the target reward distribution and $\tilde{\pi}_\theta$ is the length-normalized version of $\pi_\theta$.

Since past work has indicated that preference-tuned models often spuriously favor longer generations, we follow Malladi (2024); Meng et al. (2024) by instead using length-normalized likelihoods. Additionally, it is standard in past literature (Ziegler et al., 2019; Korbak et al., 2022a) to model $\pi^*$ as a Boltzmann function of the reward. That is,

$$\pi^*(y \mid x) = \frac{1}{Z(x)} \exp(r(x,y))$$

where $Z(x)$ is the partition function. Although this partition function is not guaranteed to converge due to the observation that LLMs often place non-zero probability mass on infinite-length sequences (Du et al., 2023; Welleck et al., 2020), we make the assumption for now that this effect is negligible. This results in a formulation of the optimal policy that is Bradley-Terry with respect to the rewards. *I.e.*,

$$\log \pi^*(y_1|x) - \log \pi^*(y_2|x) = r(x,y_1) - r(x,y_2)$$

Then, the first term of Equation 15 can be expanded as:

$$\arg \min_{\theta} \mathbb{KL}(\tilde{\pi}_\theta \| \pi^*) = \arg \min_{\theta} \mathop{\mathbb{E}}_{\substack{x \sim \mathbb{D}_x, \\ y \sim \tilde{\pi}_\theta(\cdot|x)}} \log \left( \frac{\tilde{\pi}_\theta(y|x)}{\pi^*(y|x)} \right)$$

Since we cannot directly sample from $\tilde{\pi}_\theta$, we approximate it via an expectation over the un-normalized policy:

$$\approx \arg \min_{\theta} \mathop{\mathbb{E}}_{\substack{x \sim \mathbb{D}_x, \\ y \sim \pi_\theta(\cdot|x)}} \left[ \frac{\log \pi_\theta(y|x)}{|y|} - r(x,y) + \log Z(x) \right]$$

$$= \arg \min_{\theta} \mathop{\mathbb{E}}_{\substack{x \sim \mathbb{D}_x, \\ y \sim \pi_\theta(\cdot|x)}} \left[ \frac{\log \pi_\theta(y|x)}{|y|} - r(x,y) \right]$$

Rather than re-sampling from $\pi_\theta$ after every step of training, we approximate this step using importance sampling:

$$\mathbb{KL}(\pi_\theta \| \pi^*) \approx \sum_{\substack{x \sim \mathbb{D}_x, \\ y \sim \pi_\theta(\cdot|x)}} \frac{\pi_\theta(y|x)}{\pi_{\text{Ref}}(y|x)} \pi_{\text{Ref}}(y|x) \left[ \frac{\log \pi_\theta(y|x)}{|y|} - r(x,y) \right]$$

$$\approx \mathop{\mathbb{E}}_{\substack{x \sim \mathbb{D}_x, \\ y \sim \pi_{\text{Ref}}(\cdot|x)}} \frac{\pi_\theta(y|x)}{\pi_{\text{Ref}}(y|x)} \left[ \frac{\log \pi_\theta(y|x)}{|y|} - r(x,y) \right] \tag{16}$$

For the second term of Equation 15, we remove terms not depending on $\theta$, resulting in the standard token-averaged cross-entropy loss:

$$\min_{\theta} \mathbb{KL}(\pi_{\text{Ref}} \| \pi_\theta) \equiv \min_{\theta} \mathop{\mathbb{E}}_{\substack{x \sim \mathbb{D}_x, \\ y \sim \pi_{\text{Ref}}(\cdot|x)}} - \frac{\log \pi_\theta(y|x)}{|y|} \tag{17}$$

Plugging Equations 16 and 17 back into 15, we obtain

$$\mathcal{L}_{\text{MargE}}(\pi_\theta, \pi_{\text{Ref}}; \mathbb{D}_x) = \mathop{\mathbb{E}}_{\substack{x \sim \mathbb{D}_x, \\ y \sim \pi_{\text{Ref}}(\cdot|x)}} \left[ \frac{\pi_\theta(y|x)}{\pi_{\text{Ref}}(y|x)} \left( \frac{\log \pi_\theta(y|x)}{|y|} - r(x,y) \right) - \lambda \frac{\log \pi_\theta(y|x)}{|y|} \right].$$

Since importance sampling often leads to high variance of the gradient estimates in practice (Owen, 2013), we instead use the self-normalized version of this objective (Eq. 30).

**Lemma A.1.** *Given a Bradley-Terry model $\pi(y|x) = \frac{1}{Z(x)} \exp(r(x,y))$ with partition function $Z(x)$, if reward function $r(x,y) = f(y) - f(x)$ for some real-valued scoring function $f(x) : \mathcal{V}^* \to \mathbb{R}$, then $\pi(y|x) = \pi(y|z)$ for any pair $x, z \in \text{dom}(f)$.*

*Proof.* Since $Z(x)$ is the normalizing constant for $\pi(y|x)$, we can write

$$Z(x) = \sum_{y' \in \mathcal{V}^*} \exp(r(x, y')) \tag{18}$$

$$= \sum_{y' \in \mathcal{V}^*} \exp(f(y') - f(x)) \tag{19}$$

$$= \frac{1}{\exp(f(x))} \sum_{y' \in \mathcal{V}^*} \exp(f(y')). \tag{20}$$

It follows that

$$\pi(y|x) = \frac{\exp(f(x))}{\sum_{y' \in \mathcal{V}^*} \exp(f(y'))} \exp(r(x, y)) \tag{21}$$

$$= \frac{\exp(f(x) + f(y) - f(x))}{\sum_{y' \in \mathcal{V}^*} \exp(f(y'))} \tag{22}$$

$$= \frac{\exp(f(y))}{\sum_{y' \in \mathcal{V}^*} \exp(f(y'))} \tag{23}$$

$$= \frac{\exp(f(y) + f(z) - f(z))}{\sum_{y' \in \mathcal{V}^*} \exp(f(y'))} \tag{24}$$

$$= \frac{\exp(f(z)) \exp(r(z, y))}{\sum_{y' \in \mathcal{V}^*} \exp(f(y'))} \tag{25}$$

$$= \frac{1}{Z(y)} \exp(r(z, y)) \tag{26}$$

$$= \pi(y|z) \tag{27}$$

$\square$

## A.5. Prompt

We use this prompt only in our initial prompting experiment with o1 and Gemini Flash (see Section 6.1), with <TRANSITION MATRIX HERE> replaced by the transition matrix corresponding to $f_2$.

> You are given a challenging discrete optimization problem to solve. The problem consists of generating a discrete integer sequence (with vocabulary consisting of integers from 0 to 31, inclusive) that lies in the support of a discrete Markov process (DMP) and contains specific spaced motifs. The transition matrix for the DMP is as follows:
>
> <TRANSITION MATRIX HERE>
>
> The motifs are [ 3, 16, 15, 11], [24, 3, 16, 15], [11, 14, 8, 10], [22, 27, 7, 20] and the respective spacings for these motifs are [0, 2, 4, 5], [0, 3, 5, 6], [0, 1, 4, 6], and [0, 2, 4, 15]. Your solution will be scored based both on whether it is feasible (i.e. lies in the support of the DMP) and how many motifs it fulfills. That is, if the $c$ motifs and their spacings are denoted as $\{(m^{(1)}, s^{(1)}), \cdots, (m^{(c)}, s^{(c)})\}$, then we score whether sequence $x$ fulfills motif $i$ with $h(x, m^{(i)}, s^{(i)}) = \max_{l < L}(\sum_{j=1}^{k} \mathbb{1}[x_{l+s_j^{(i)}} = m_j^{(i)}])/4$ where $L$ is the length of $x$ ($L = 32$ in this problem). If the sequence $x$ is feasible, then the total score is $\prod_{i=1}^{c} h(x, m^{(i)}, s^{(i)})$. Otherwise, the score is 0.
>
> Here are some example sequences and their respective scores:
> $x_1 = [12, 31, 2, 4, 15, 7, 14, 15, 12, 31, 11, 29, 25, 1, 15, 11, 19, 24, 22, 5, 17, 27, 1, 14, 31, 28, 16, 15, 11, 14, 16, 10]$
> Score of $x_1$: $(1/4) * (1/4) * (1/2) * (1/2) = 1/64$
>
> $x_2 = [12, 31, 2, 4, 15, 11, 14, 15, 12, 31, 11, 10, 25, 1, 15, 11, 19, 24, 22, 5, 17, 27, 1, 14, 31, 28, 16, 15, 11, 14, 10, 15]$
> Score of $x_2$: 0
>
> $x_3 = [3, 16, 15, 11, 24, 24, 24, 24, 15, 11, 22, 22, 22, 22, 22, 22, 22, 22, 22, 22, 22, 22, 22, 22, 22, 22, 22, 22, 22, 22, 22, 22]$
> Score of $x_3$: $(1/2) * (1/4) * (1/4) * (1/4) = 1/128$
>
> $x_4 = [3, 3, 16, 16, 15, 11, 24, 24, 24, 24, 15, 11, 22, 22, 22, 22, 22, 22, 22, 22, 22, 22, 22, 22, 22, 22, 22, 22, 22, 22, 22, 22]$
> Score of $x_4$: $1 * (1/4) * (1/4) * (1/4) = 1/64$
>
> It is often a good idea to start with a random sequence, perturb it a bit, score it and stay with this perturbed version if it scores better. You can repeat this process as many times as you want until the score converges to the maximum value. It is extremely important to check your progress once in a while. If you haven't made good progress, it would be a good idea to backtrack what you have done (make sure to mark important steps with special markers so that you know where to backtrack to) and resume from there on.
>
> Since there could be many isolated solutions, it would be a good idea to try this process from a few different random initial sequences. Make sure to score the solution of each of these random initial sequences and revisit them at the end to find the very best sequence.
>
> Everyone's life depends on finding the best sequence, as this sequence would be a key to finding a medicine for the future pandemic.
>
> It is important for you to solve this problem directly yourself by thinking about it carefully and iteratively, as you are not allowed to use the computer, due to safety measures. Please directly output a sequence and not code, as you do not have access to an environment to execute the code in.

## A.6. Algorithms

---

**Algorithm 2** DATASETFORMATTING($S$). An algorithm that adapts PropEn (Tagasovska et al., 2024) to create pairs or triples of data for LLM training. We use $k_n = 30$, $\Delta x = 0.25$, and the fractional Hamming distance as $d$.

---

**Input:** Dataset $S = \{(\mathbf{x}, y)\}$ of particles $\mathbf{x}$ and scores $y$; $k_n$ number of nearest neighbors to find; distance function $d$; threshold $\Delta x$; dataset type $type = $ ("binary" | "triple")

**if** $type = $ *"binary"* **then**

$$\mathcal{D} \leftarrow \left\{ (x_i, x_j) \,\middle|\, \begin{array}{c} (x_i, y_i),(x_j,y_j)\in S \\ x_j \in \text{KNEARESTNEIGHBORS}(x_i, k_n) \\ d(x_i, x_j) \le \Delta_x \\ f(x_j) > f(x_i) \end{array} \right\}$$

**else**

$$\mathcal{D} \leftarrow \left\{ (x_i, x_j, x_k) \,\middle|\, \begin{array}{c} (x_i, y_i),(x_j,y_j),(x_k,y_k)\in S \\ x_j, x_k \in \text{KNEARESTNEIGHBORS}(x_i, k_n) \\ d(x_i, x_j) \le \Delta_x, d(x_i, x_k) \le \Delta_x \\ f(x_j) > f(x_i), f(x_i) \ge f(x_k) \end{array} \right\}$$

**end**

**Output:** $\mathcal{D}$

---

**Algorithm 3** ITERATIVEREFINEMENT($\pi_\theta$; $S$). $\pi_\theta^t$ represents $\pi_\theta$ with temperature $t$ scaling. We use $n_s = 200$, $n_i = 10$, $n_o = 10$, and $T = [0.6, 0.8, 1.0, 1.2, 1.4, 1.6]$.

---

**Input:** Pretrained LLM $\pi_\theta$; dataset $S = \{(\mathbf{x}, y)\}$ of sequences $\mathbf{x}$ and scores $y$; $n_s$ number of seed examples; $n_i$ rounds of iteration per example; $n_o$ outputs per iteration; $T$ set of temperatures to sample with.

$S' \leftarrow \{\}$
$\mathcal{X} \leftarrow \{\mathbf{x} \mid (\mathbf{x}, y) \in \text{TOPK}(S, n_s)\}$          ▷ Obtain $n_s$ seed examples from the top training examples by score.
**for** $\mathbf{x} \in \mathcal{X}$ **do**
    $\mathbf{x}_0 \leftarrow \mathbf{x}$
    $i \leftarrow 0$
    **for** $i < n_i$ **do**
        $\mathbf{x}_{i+1} \leftarrow \text{GREEDYDECODING}(\pi_\theta; \mathbf{x}_i)$
        $S' \leftarrow S' \cup \{\mathbf{x}_{i+1}\}$
        $i \leftarrow i + 1$
    **end**
    **for** $t \in T$ **do**
        $i \leftarrow 0$
        **for** $i < n_i$ **do**
            $\mathcal{X}_i \leftarrow \{\mathbf{x}_j \sim \pi_\theta^t(\cdot \mid \mathbf{x}_i)\}_{j=1}^{n_o}$      ▷ Sample $n_o$ sequences using temperature $t$.
            $S' \leftarrow S' \cup \mathcal{X}_i$
            $\mathbf{x}_{i+1} \leftarrow \arg\max_{\mathbf{x}' \in \mathcal{X}_i} \pi_\theta(\mathbf{x}' \mid \mathbf{x})$      ▷ Select the highest-likelihood sample as the input for the next iteration.
            $i \leftarrow i + 1$
        **end**
    **end**
**end**
$S' \leftarrow \text{DEDUPLICATE}(S')$
**Output:** $S'$

---

**Algorithm 4** FILTER($\mathcal{X}, j$). An algorithm for filtering a dataset $\mathcal{X}$ of sequences down to only $j$ sequences.

---

**Input:** LLM $\pi_\theta$ that generated the sequences; dataset $\mathcal{X} = \{(\mathbf{x}, y)\}$ of sequences $\mathbf{x}$ and scores $y$; likelihood threshold $p_{\min}$; maximum proportion of infeasible sequences $p_{\text{max-infeas}}$; final output size $k$.

$\mathcal{X} \leftarrow \{((\mathbf{x}, y), \pi_\theta(\mathbf{x})) \mid (\mathbf{x}, y) \in \mathcal{X}\}$      ▷ Compute likelihoods.
$\mathcal{X} \leftarrow \{(\mathbf{x}, y) \mid \pi_\theta(\mathbf{x}) > p_{\min}, ((\mathbf{x}, y), \pi_\theta(\mathbf{x})) \in \mathcal{X}\}$
$\mathcal{X}_{\text{feasible}} \leftarrow \{(\mathbf{x}, y) \mid y \ne -\infty, (\mathbf{x}, y) \in \mathcal{X}\}$
$\mathcal{X}_{\text{infeasible}} \leftarrow \{(\mathbf{x}, y) \mid y = -\infty, (\mathbf{x}, y) \in \mathcal{X}\}$
$n_{\text{feasible}} \leftarrow |\mathcal{X}_{\text{feasible}}|$
$n_{\text{max-infeas.}} \leftarrow n_{\text{feasible}} \times \frac{p_{\text{max-infeas}}}{1 - p_{\text{max-infeas}}}$
$\mathcal{X} \leftarrow \mathcal{X}_{\text{feasible}} \cup \text{SAMPLE}(\mathcal{X}_{\text{infeasible}}, n_{\text{max-infeas.}})$      ▷ Downsample infeasible examples.
$\mathcal{X} \leftarrow \text{SAMPLE}(\mathcal{X}, \min(j, |\mathcal{X}|))$      ▷ Subsample $j$ examples.
**Output:** $\mathcal{X}$

---

### A.6.1. DATASET FORMATTING

To format our data, we adapt PropEn (Tagasovska et al., 2024), a technique for matching pairs of examples to implicitly guide the model towards generating sequences that are close by the input but still improve a particular property. Since PropEn was originally designed for pairs of data $(x, y)$, we also adapt PropEn to create triples of data $(x, y_w, y_l)$ for DPO training. In short, given a dataset $\mathcal{D} = \{(x_i, y_i)\}_{i=1}^N$ of input sequences $x_i$ and their scores $y_i$, PropEn creates the following paired dataset:

$$\mathcal{D}_{\text{PropEn}} = \{(x_i, x_j) \mid (x_i, y_i), (x_j, y_j) \in \mathcal{D}, d(x_i, x_j) \leq \Delta_x, f(x_j) - f(x_i) \in (0, \Delta_y]\} \tag{28}$$

for thresholds $\Delta_x$ and $\Delta_y$ and distance function $d$. In all our experiments, we use the Hamming distance as $d$ and modify the constraint $f(x') - f(x) \in (0, \Delta_y)$ to $f(x') > f(x)$ since we observed that this looser constraint was more effective in our experiments. We use $\mathcal{D}_{\text{PropEn}}$ for both LLOME-SFT and LLOME-MARGE.

To additionally adapt the PropEn dataset creation process for preference tuning (*e.g.* DPO), we create preference triples using the following constraints:

$$\mathcal{D}_{\text{PropEn-Triples}} = \left\{ (x_i, x_j, x_k) \middle| \begin{array}{l} (x_i, y_i), (x_j, y_j), (x_k, y_k) \in \mathcal{D} \\ d(x_i, x_j) \leq \Delta_x, d(x_i, x_k) \leq \Delta_x \\ f(x_j) > f(x_i), f(x_i) \geq f(x_k) \end{array} \right\} \tag{29}$$

In preference tuning terms, the $x_i$ is the prompt or input sequence, and $x_j$ and $x_k$ are $y_w$ and $y_l$, respectively. We use $\mathcal{D}_{\text{PropEn-Triples}}$ in LLOME-DPO. We formalize this algorithm in Alg. 2.

For all training algorithms, we format the input as "`<inc> [3, 1, ⋯, 5]`" where "`<inc>`" is a control code meant to indicate to the model that it should edit the sequence to increase the score. Outputs are formatted similarly, but without the control code.

### A.6.2. ITERATIVE REFINEMENT

Our iterative refinement is formalized in Alg. 3. Loosely, we select the best $n_s$ training examples from the last round of the LLOME outer loop, and provide them as seed inputs to the LLM to refine. For each seed input, we use 10 rounds of iterative generation where the best (highest-likelihood) generation from the previous round is provided as input to the next round. We repeat this process with both greedy decoding and sampling at various temperatures. In the case of greedy decoding, only one generation is obtained per iteration. When we sample, we sample $n_o = 10$ outputs at once.

Since LLM outputs tend to become less diverse with more rounds of training, we also implement automatic temperature adjustments after each round of LLOME. The default temperature range is $T = [0.6, 0.8, 1.0, 1.2, 1.4, 1.6]$, but if the average Hamming distance of generations from the last LLOME round was $< 0.075$, then we instead use $T + 0.6$. For average Hamming distance between $0.075$ and $0.1$, we use $T + 0.4$. For averages between $0.1$ and $0.1$, we use $T + 0.2$.

### A.6.3. GENETIC ALGORITHM

In Algorithms 5, 6, and 7, we provide pseudo-code for our genetic algorithm baseline, which we implement in pure PyTorch (Paszke et al., 2019), using the `torch.optim` API.

The GA baseline has only four hyperparameters, the total number of particles $n$, the survival quantile $\alpha \in (2/n, 1)$, the mutation probability $p_m$, and the recombination probability $p_r$. Generally speaking for best performance one should use the largest $n$ possible, and tune $\alpha$ (which determines the greediness of the optimizer), $p_m$, and $p_r$.

Our genetic algorithm uses mutation probability $p_m = 0.005$, $n = 1000$ particles per iteration, survival quantile $\alpha = 0.1$, and recombination probability $p_r = 0.0882$.

---

**Algorithm 5** Genetic algorithm pseudo-code

---

**Input:** initial solution $\widehat{\mathbf{x}^*}, \widehat{f^*}$, mutation probability $p_m$, recombination probability $p_r$, survival quantile $\alpha$, # particles $n$

$\mathcal{X}_{\text{pop}} \leftarrow \texttt{mutate}(\{\widehat{\mathbf{x}^*}\}, p_m, n)$

**for** $t = 1, \ldots, T$ **do**

    $\mathbf{v} \leftarrow f(\mathcal{X}_{\text{pop}})$

    **if** $\max v_i > \widehat{f^*}$ **then**

        $\widehat{\mathbf{x}^*} \leftarrow \arg\max v_i$

        $\widehat{f^*} \leftarrow \max v_i$

    **end**

    $\tau \leftarrow \texttt{quantile}(\mathbf{v}, 1 - \alpha)$

    $\mathcal{X}_{\text{top}} \leftarrow \{\mathbf{x} \in \mathcal{X}_{\text{pop}} \mid f(\mathbf{x}) \geq \tau\}$

    $n' \leftarrow n - |\mathcal{X}_{\text{top}}|$

    $\mathcal{X}_{\text{pop}} \leftarrow \mathcal{X}_{\text{top}} \cup \texttt{recombine}(\mathcal{X}_{\text{top}}, p_r, n')$

    $\mathcal{X}_{\text{pop}} \leftarrow \texttt{mutate}(\mathcal{X}_{\text{pop}}, p_m, 1)$

**end**

**Returns:** Estimated maximizer $\widehat{\mathbf{x}^*}, \widehat{f^*}$

---

**Algorithm 6** `mutate` function

---

**Input:** initial set $\mathcal{X}$, mutation probability $p_m$, number of mutants $n$.

$\mathcal{X}' = \emptyset$

**for** $\mathbf{x} \in \mathcal{X}$ **do**

    **for** $i = 1, \ldots, n$ **do**

        $\texttt{mask} = \texttt{rand\_like}(\mathbf{x}) < p_m$

        $\texttt{sub} = \texttt{randint}(0, v - 1, \texttt{len}(\mathbf{x}))$

        $\mathbf{x}' = \texttt{where}(\texttt{mask}, \texttt{sub}, \mathbf{x})$

        $\mathcal{X}' = \mathcal{X}' \cup \{\mathbf{x}'\}$

    **end**

**end**

**Returns:** $\mathcal{X}'$

---

**Algorithm 7** `recombine` function

---

**Input:** initial set $\mathcal{X}$, recombine probability $p_r$, number of recombinations $n$.

$\mathcal{X}' = \emptyset$

$\mathcal{P}^{(1)} = \texttt{draw\_w\_replacement}(\mathcal{X}, n)$

$\mathcal{P}^{(2)} = \texttt{draw\_w\_replacement}(\mathcal{X}, n)$

**for** $i = 1, \ldots, n$ **do**

    $\mathbf{x}^{(1)} = \mathcal{P}^{(1)}_i$

    $\mathbf{x}^{(2)} = \mathcal{P}^{(2)}_i$

    $\texttt{mask} = \texttt{rand\_like}(\mathbf{x}^{(1)}) < p_r$

    $\mathbf{x}' = \texttt{where}(\texttt{mask}, \mathbf{x}^{(1)}, \mathbf{x}^{(2)})$

    $\mathcal{X}' = \mathcal{X}' \cup \{\mathbf{x}'\}$

**end**

**Returns:** $\mathcal{X}'$

---

## A.7. LLM Training Details

We train every model for 1 epoch with PyTorch DDP on two A100 GPUs, using training loops implemented with the Huggingface `datasets`, `transformers`, and `trl` libraries. We conducted hyperparameter searches for each LLM training method, using the validation loss from the dataset of the first iteration of LLOME to select the best hyperparameters. We also check whether the generated outputs are parsable and conform to the correct format (*i.e.*, a list of the correct length with values in the correct range). If the hyperparameter set-up with the lowest validation loss does not output sequences with the correct format $> 90\%$ of the time, then we select the set-up with the next best validation loss that meets these constraints. Notably, these format checks were the most important for DPO. Many DPO-trained models achieve low validation loss despite generating sequences with incorrect format. We use the best hyperparameters tuned from the validation dataset created during the first iteration of LLOME but do not repeat hyperparameter tuning in future iterations. All search ranges and final hyperparameter values (in **bold**) are listed below.

**SFT**    We train the SFT models with the AdamW optimizer, with $\beta_1 = 0.9$, $\beta_2 = 0.999$, $\lambda = 0.01$, and $\epsilon = 1 \times 10^{-8}$. We also search the following hyperparameter ranges:

- Learning rate $\in \{1 \times 10^{-7}, \mathbf{1 \times 10^{-6}}, 1 \times 10^{-5}\}$
- Batch size $\in \{16, 32, 64, \mathbf{128}\}$

**DPO**    We train the DPO models with the RMSprop optimizer, with $\alpha = 0.99$, $\lambda = \mu = 0$, and $\epsilon = 1 \times 10^{-8}$. Due to computational constraints, we train with `bf16`. We also search the following hyperparameter ranges:

- Learning rate $\in \{\mathbf{1 \times 10^{-7}}, 1 \times 10^{-6}\}$
- Batch size $\in \{\mathbf{64}, 128\}$
- $\beta \in \{0.1, 0.2, 0.4, 0.8\}$

**MargE**    We train the MargE models with the AdamW optimizer, with $\beta_1 = 0.9$, $\beta_2 = 0.999$, $\lambda = 0.01$, and $\epsilon = 1 \times 10^{-8}$. We also search the following hyperparameter ranges:

- Learning rate $\in \{1 \times 10^{-7}, \mathbf{1 \times 10^{-6}}\}$
- Batch size $\in \{\mathbf{64}, 128\}$
- $\lambda \in \{0.2, 0.4, 0.8, 1.0, \mathbf{10.0}\}$

**REINFORCE**    We trained REINFORCE with the same best hyperparameters as MargE.

**Self-Normalization**    For both MargE and REINFORCE, we applied self-normalization to the importance weights. That is, if $\mathcal{B}(x, y)$ is the batch of examples that a particular example $(x, y)$ belongs to, then the self-normalized MargE and REINFORCE objectives are as follows:

$$\tilde{\mathcal{L}}_{\text{MargE}}(\pi_\theta, \pi_{\text{Ref}}; \mathbb{D}_x) = \mathop{\mathbb{E}}_{\substack{x \sim \mathbb{D}_x, \\ y \sim \pi_{\text{Ref}}(\cdot|x)}} \left[ \tilde{w}(x, y) \left( \frac{\log \pi_\theta(y|x)}{|y|} - r(x, y) \right) - \lambda \frac{\log \pi_\theta(y|x)}{|y|} \right] \tag{30}$$

$$\tilde{\mathcal{L}}_{\text{REINFORCE}}(\pi_\theta, \pi_{\text{Ref}}; \mathbb{D}_x) = \mathop{\mathbb{E}}_{\substack{x \sim \mathbb{D}_x, \\ y \sim \pi_{\text{Ref}}(\cdot|x)}} \left[ \tilde{w}(x, y) \left( -r(x, y) \log \pi_\theta(y|x) \right) - \lambda \frac{\log \pi_\theta(y|x)}{|y|} \right] \tag{31}$$

where

$$w(x, y) = \pi_\theta(y|x)/\pi_{\text{Ref}}(y|x), \tag{32}$$

$$\tilde{w}(x, y) = \frac{w(x, y)}{\sum_{(x', y') \in \mathcal{B}(x, y)} w(x', y')}. \tag{33}$$

**Updating $\pi_{\text{Ref}}$ During Iterative Training**    In Algorithm 1, we iteratively train the LLM using outputs generated by the last round of iterative refinement. In this iterative training process, at iteration $i$ of the outer loop, we always update $\pi_{\text{Ref}}$ for DPO, MargE, and REINFORCE such that $\pi_{\text{Ref}} := \pi_{\theta_i}$.

## A.8. LaMBO-2 Model and Training Details

We built upon the implementations of LaMBO-2 in cortex and poli-baselines. We use a LaMBO-2 architecture with an encoder that is shared between different output modules: one generative discrete diffusion head, one discriminative head predicting whether a sequence satisfies the constraints of the problem, and one discriminative head predicting the reward of a sequence.

Each of these modules is made up of 1D CNN residual blocks, with layer normalization and sinusoidal position embeddings (except for the generative head, which is a linear layer directly from the shared embeddings to output logits over the vocabulary). The encoder is composed of 2 residual blocks and the two discriminative heads are composed of 1 residual block each, all with kernel width 5. Each residual block has 128 channel dimensions and 128 embed dimensions. We applied diffusion noise and guidance to the encoder embedding, and for each of the discriminative tasks, we trained an ensemble of 8 independently randomly initialized heads.

### A.8.1. TRAINING SET REBALANCING

We employ a data rebalancing strategy during the training of LaMBO-2 to enhance continual learning. The objective is to enable the model to adapt effectively to newly acquired data while preserving knowledge learned from historical data. This is achieved by a sampling mechanism that effectively gives exponentially greater importance to more recent data during training.

**Partitioning Scheme:**   Training data is partitioned based on the iteration in which it was collected. This partitioning follows a geometric scheme:

- Partition 0 contains data from the most recent iteration.

- Partition 1 contains data from the two iterations immediately preceding those in Partition 0.

- Partition 2 contains data from the four iterations immediately preceding those in Partition 1.

- This pattern continues, with Partition $k$ generally containing data from $2^k$ distinct iterations that are older than the data in Partition $k-1$.

Formally, let $T$ be the index of the most recent data collection iteration. Data collected during iteration $i$ (where $i \leq T$) is assigned to partition $p(i)$ according to:

$$p(i) = \begin{cases} 0 & \text{if } i = T, \\ k & \text{if } i \in [T - 2^{k+1} + 2, T - 2^k + 1] \text{ for } k \geq 1. \end{cases}$$

Each partition $k$ (for $k \geq 1$) thus groups data from $2^k$ consecutive iterations.

**Round-Robin Partition Rebalancing Sampling Mechanism:**   During training, we construct minibatches by sampling one datapoint from each active partition. Since more recent data is grouped into smaller partitions (e.g., Partition 0 contains data from only one iteration, while Partition $k$ contains data from $2^k$ iterations), datapoints from recent partitions have a higher probability of being selected. This sampling strategy naturally creates the desired exponential weighting, prioritizing recent information.

To prepare data for training:

- Datapoints within each individual partition are randomly shuffled.

- The order in which the partitions themselves are processed for sampling is also randomized. This shuffled order defines a cycle for drawing samples.

To form a training minibatch of a predefined size:

- We iterate through the partitions according to their current shuffled order.

- From each partition encountered in this cycle, one (randomly selected) datapoint is drawn and added to the current minibatch.

- This process continues, drawing one datapoint from each subsequent partition in the shuffled order, until the minibatch reaches its target size.

- If all partitions have been sampled from (i.e., one cycle through the shuffled partition order is complete) and the minibatch is not yet full, the order of partitions is re-shuffled, and sampling continues from the newly ordered partitions until the minibatch is filled.

- When any partition is exhausted, the examples in that partition are shuffled and sampling continues.

### A.8.2. CONTROLLING DIVERSITY THROUGH CANDIDATE SELECTION

We also introduce a new hyperparameter to LaMBO-2, the "farthest first traversal (FFT) expansion factor", which provides a lever to control the diversity of solutions used as starting points for guided diffusion. For applications with a similar number of ground truth evaluations $n$ per evaluation iteration compared to the total number of iterations (e.g. a 24-well plate of samples for each of 10 iterations), the best FFT expansion factor $\alpha$ is usually greater than one. In this setting, we retrieve the top $\alpha n$ solutions seen over the optimizer's history and use farthest first traversal to select the subset of $n$ solutions which capture as much sequence diversity as possible among the original $\alpha n$ solutions. The FFT procedure iteratively adds a solution to the selected subset if it is maximally distant from all the currently selected solutions (See Algorithm 9 for pseudo-code). In this small $n$ setting, $\alpha > 1$ prevents generator collapse and trades exploitation for more exploration.

In contrast, when $n$ is very large compared to the number of iterations (such as in the experiments for this paper, with $n = 2000$ and only 10 iterations), it is best to set $\alpha < 1$. In this case, we select the top $\alpha n$ solutions from the optimizer's history to seed the next round, repeating them as needed to have exactly $n$ seeds. Since the guided diffusion is not deterministic, despite seeding multiple generative trajectories from the same starting sequence, the model still generates solution pools with high diversity. See Algorithm 8 for pseudo-code of our candidate starting point selection strategy, covering both the $\alpha < 1$ and $\alpha > 1$ settings.

---

**Algorithm 8** Get Candidate Starting Points from LaMBO-2 Solution History

---

**Input:** History of solutions $x$, scores $y$, FFT expansion factor $\alpha$, number of samples to evaluate this iteration $n$, edit distance function `edit_dist`

`sorted_indices` $\leftarrow$ argsort($y$, descending=True)
$K \leftarrow \min(\text{len}(x), \alpha n)$
`candidate_points` $\leftarrow x[\text{sorted\_indices}[: K]]$
**if** $\alpha > 1$ **then**
    `indices` $\leftarrow$ FarthestFirstTraversal(`candidate_points`, `edit_dist`, `candidate_scores`, $n$)

**else**
    `repeat_factor` $\leftarrow \lfloor 1/\alpha \rfloor + 1$
    `candidate_points` $\leftarrow$ repeat(`candidate_points`, `repeat_factor`)$[: n]$
    `indices` $\leftarrow$ arange(len(`candidate_points`))

**end**
**Returns:** `candidate_points[indices]`

---

---

**Algorithm 9** Farthest First Traversal

---

**Input:** Library $L$ of $N$ elements, distance function $d$, ranking scores $s$, number of elements to select $n$

$R \leftarrow \text{argsort}(s)$ // Sort indices by scores
$S \leftarrow [R[0]]$ // Initialize selected indices with first element
$R \leftarrow R[1:]$ // Remove first element from remaining
$PQ \leftarrow \emptyset$ // Initialize priority queue
**for** $i = 0, \ldots, |R| - 1$ **do**
  dist $\leftarrow -d(L[i], L[S[0]])$ // Negative distance for max-heap
  $PQ.\text{push}((\text{dist}, s[i], i, 1))$
**end**
**for** $i \leftarrow 1$ **to** $n - 1$ **do**
  **while** *True* **do**
    $(dist, score, idx, checked) \leftarrow PQ.\text{pop}()$
      **if** $checked < |S|$ **then**
        $min\_dist \leftarrow \min(d(L[idx], L[S[j]])$ for $j \in [checked, |S|)$
        $min\_dist \leftarrow \min(min\_dist, -dist)$
        $PQ.\text{push}((-min\_dist, score, idx, |S|))$
      **else**
        $S.\text{append}(idx)$
          **break**
      **end**
  **end**
**end**
**Returns:** $S$

---

### A.8.3. TRAINING AND GENERATION HYPERPARAMETER SWEEPS

We trained LaMBO-2 with a batch size of 128, using the Adam optimizer with $\beta_1 = 0.9, \beta_2 = 0.99, \gamma = 0.005$ and no weight decay. We also searched over the following hyperparameter ranges:

- Number of design steps per iteration $\in \{8, 16, \mathbf{32}\}$

- Number of mutations per design step $\in \{2, \mathbf{4}, 8, 16\}$

- Diffusion guidance step size $\in \{0.01, \mathbf{0.05}, 0.1, 0.2\}$

- Training epochs $\in \{8, 16, 50, \mathbf{100}\}$

- Farthest first traversal (FFT) expansion factor $\in \{0.1, \mathbf{0.25}, 0.5, 1.0\}$

## A.9. Additional Results

**How sensitive is each method to hyperparameters, test function difficulty, and seed dataset?** An important aspect of an optimization algorithm is how robust it is to hyperparameter settings, problem difficulty, and choice of seed examples. To explore robustness, we present the Pareto frontiers and corresponding hypervolumes of all methods across all test functions for evaluation budgets ranging from 1K to 30K in Fig. 10. All three LLOME variants exhibit significantly lower average hypervolume than both the GA and LAMBO-2.

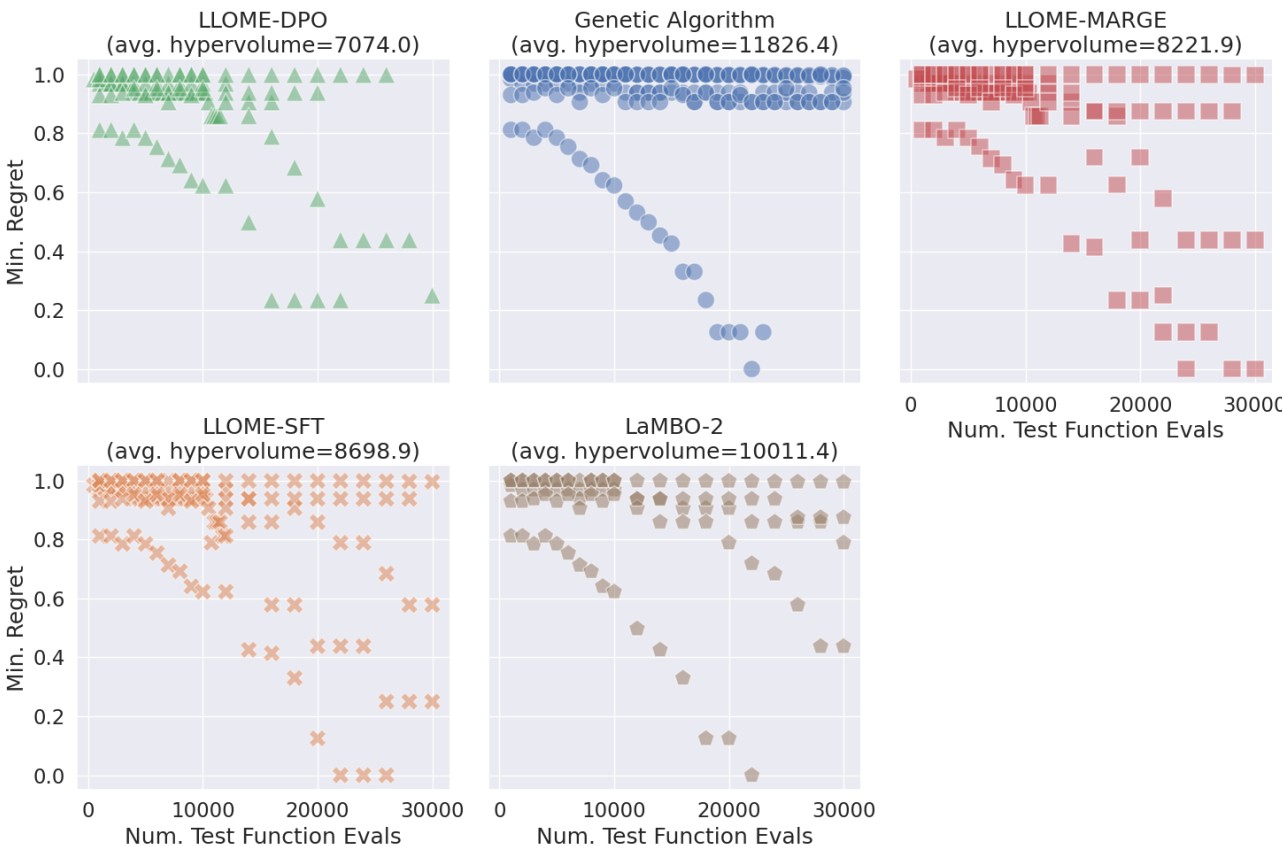

Figure 10: Pareto frontiers of evaluation budget vs. minimum regret for a variety of LLOME and LAMBO-2 hyperparameter settings, Ehrlich functions, and seed datasets. The average hypervolume refers to the average (number of test function evaluations × minimum regret). Lower hypervolume is better. However, the average hypervolume of LLOME-DPO is artificially deflated due to many DPO experiments ending prematurely as a result of generator collapse.

**How sensitive is LLOME to model size?** We compare LAMBO-2 against LLOME-MARGE with a smaller LLM ( 226K params, based on the LLaMA architecture), trained from scratch, since the Pythia model used in our main results (Fig. 3) is much larger in model size (2.8B parameters versus 314K) and has been pre-trained on the Pile (Gao et al., 2020). Although pre-training should not offer any additional advantages due to the lack of overlap between Ehrlich functions and the pre-training data, we choose this setting to be similar in model size and training to LaMBO-2. We evaluate on the $f_2$ function (i.e. **Ehr(32, 32)-4-4-4**). Results are shown in Fig. 11. Although LLOME-MARGE (LLAMA 226K) performs comparably to LLOME-MARGE (PYTHIA 2.8B) and LAMBO-2 up to 22K test function evaluations, its performance eventually plateaus, perhaps owing to limited model capacity. Additionally, the 226K model often exhibits significant training instability, including numerous spikes in both loss and gradient norm. This instability may be due to the lack of pre-training.

**At inference time, LLMs can iteratively extrapolate beyond their training distributions.** Extrapolating beyond the training distribution is a well-known machine learning problem (Bommasani et al., 2021; Press et al., 2022; Li et al., 2024),

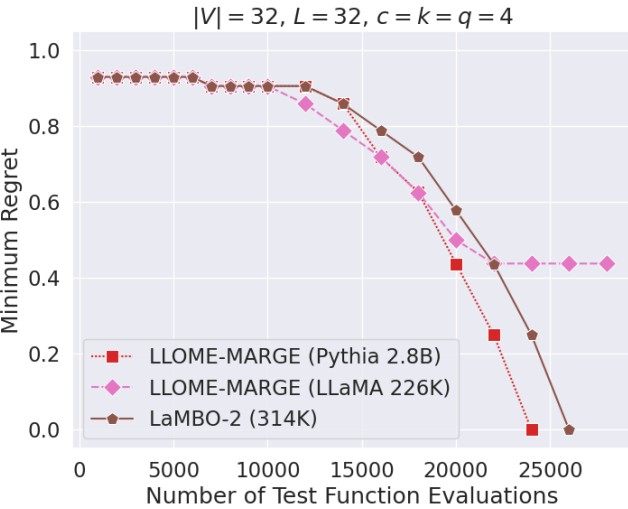

Figure 11: Minimum regret achieved by LLOME-MARGE with two LLMs of different sizes, compared against LAMBO-2 on test function $f_2$.

especially without explicit guidance provided at inference time. Although some prior work has shown LLMs to be effective at iteratively generating sequences that monotonically increase a particular attribute to values beyond the training distribution (Chan et al., 2021; Padmakumar et al., 2023), much of this work focuses on simpler tasks such as increasing the positive sentiment of text or decreasing the $\Delta\Delta G$ of a well-studied protein. Since optimizing an Ehrlich function requires satisfying multiple constraints in addition to generating sequences that lie within a small feasible region, we posit that this evaluation is a more challenging assessment of LLMs' inference-time extrapolation capabilities.

We display the iterative refinement results of LLOME's inner loop in Fig. 16, which suggest that LLMs iteratively produce edits that significantly reduce regret. However, the first few edits frequently improve the sequence whereas later edits are less likely to be helpful. This suggests that LLM's inference-time extrapolative capabilities are limited – without further training or explicit guidance, LLMs may be unable to continuously improve a given sequence beyond a certain threshold. By alternating between optimizing the model's parameters and optimizing the model's outputs, we provide a sample-efficient method for iteratively bootstrapping the model's extrapolative abilities using its own generations.

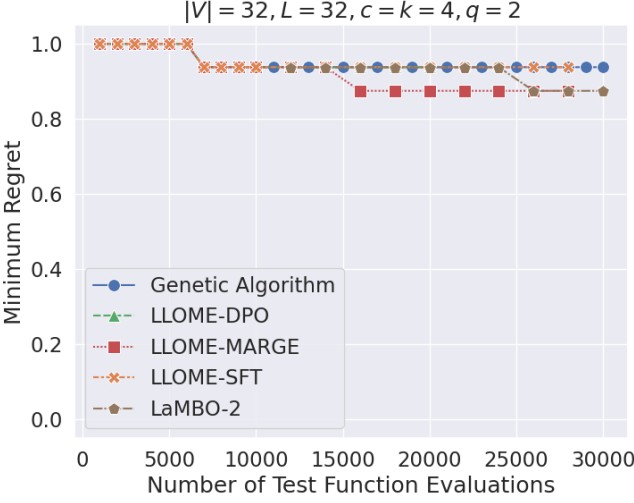

Figure 12: Minimum regret achieved as a function of the number of test function evaluations, on a test function similar to $f_2$ but with half quantization.

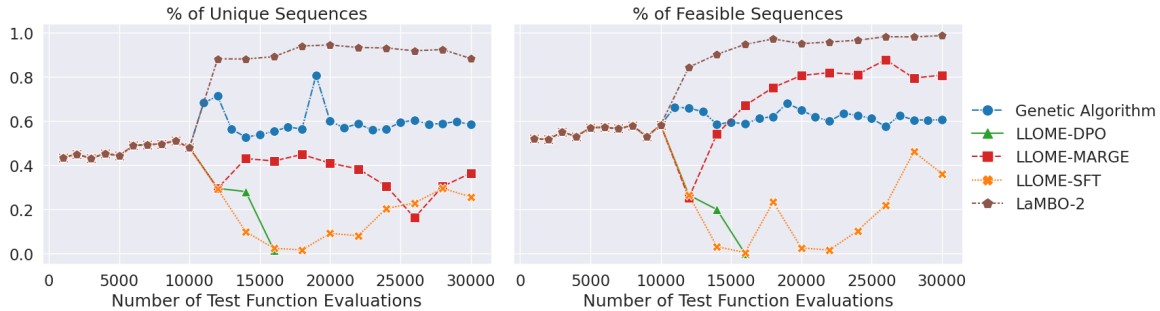

Figure 13: The percentage of generated sequences for $f_3$ that are unique or feasible. The line for LLOME-DPO ends early due to degeneration of solutions.

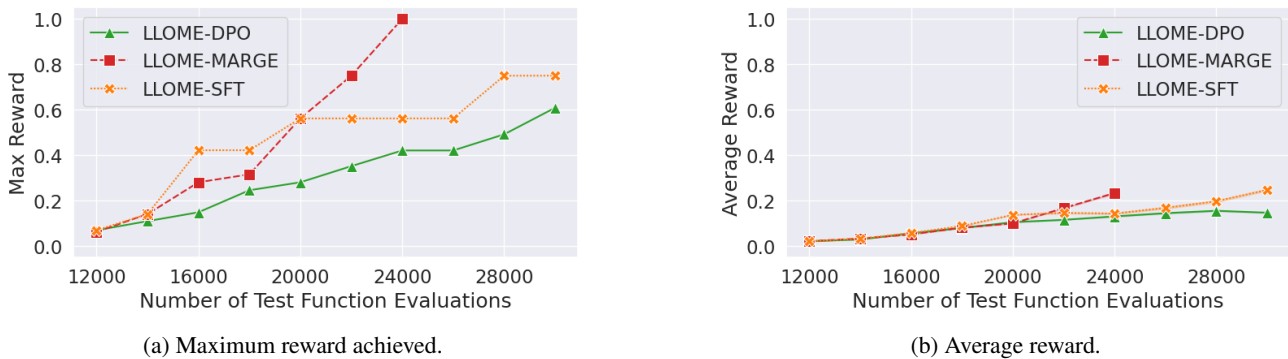

(a) Maximum reward achieved.

(b) Average reward.

Figure 14: Average and maximum reward achieved by methods that rely upon editing the original sequence. The shaded regions in (14b) represent the 95% confidence interval. The lines for LLOME-MARGE end early because LLOME-MARGE discovers the optimal solution early.

In Figure 14, we show the average and maximum reward achieved by each LLOME variant on $f_2$. While all three variants achieve similar maximum reward throughout all iterations, LLOME-SFT and LLOME-MARGE achieve significantly higher average reward than LLOME-DPO.

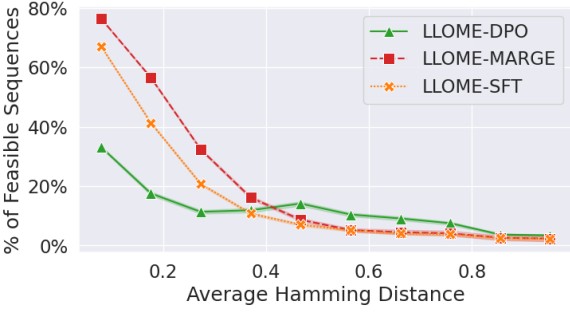

Figure 15: The percentage of feasible LLM-generated sequences, binned by the average Hamming distance (normalized by length) between the input and output. Shaded regions indicate the 95% confidence interval.

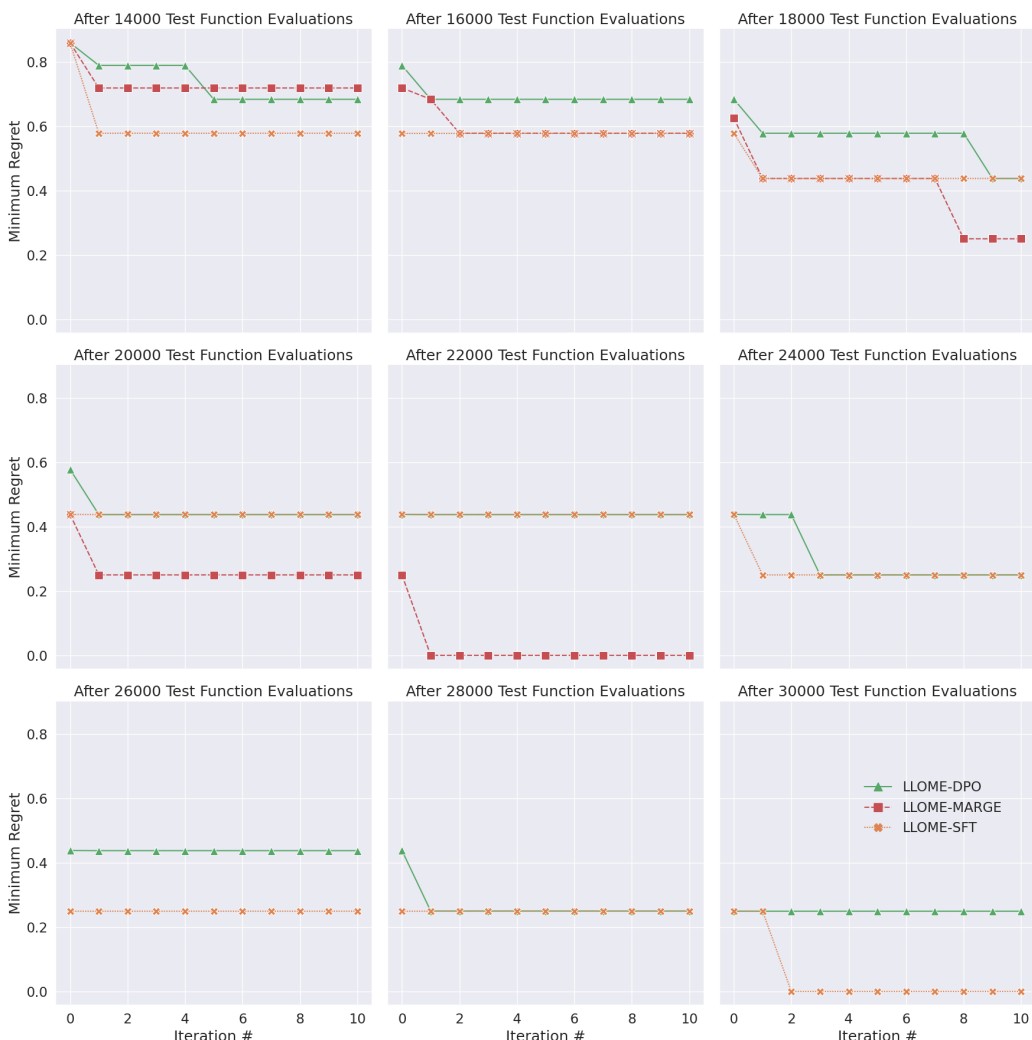

Figure 16: Minimum regret of sequences generated during the LLM inner loop, at each iteration of the iterative refinement process. The titles reflect the number of oracle labels that each LLM has been trained on. These plots account for all generations sampled from the LLM inner loop, and not just the samples selected via likelihood selection, as in Alg. 3.

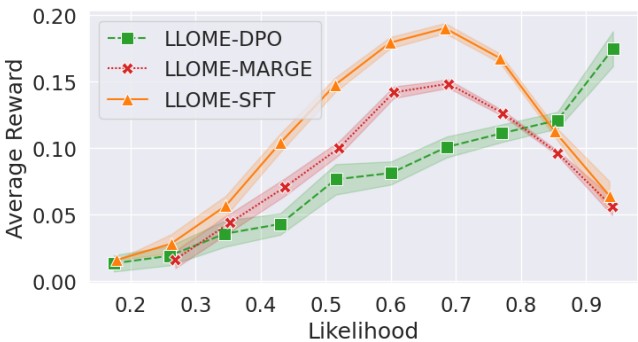

Figure 17: Likelihood vs. reward.

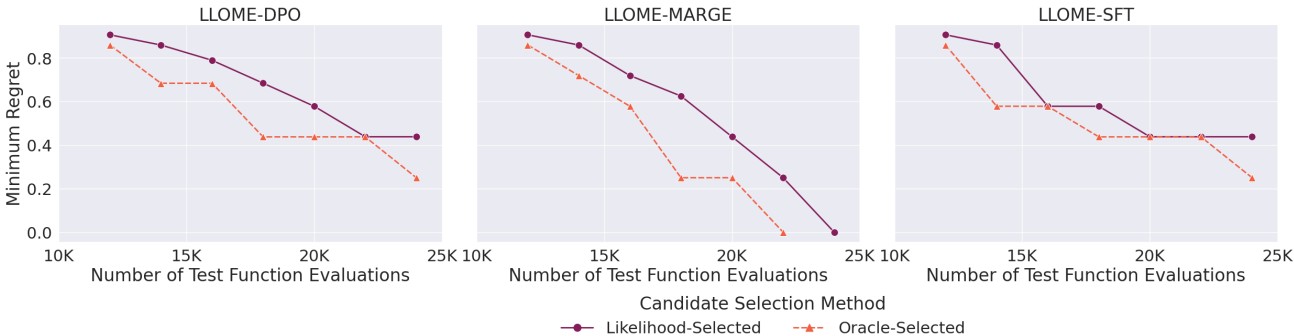

Figure 18: Minimum regret of candidates selected by either LLM likelihood or the oracle on the $f_2$ test function. Since the first 10K test function evaluations are seeds derived from the genetic algorithm, we show only candidates generated after the first 10K.

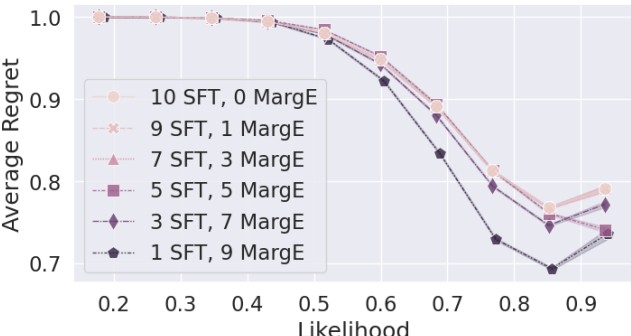

Figure 19: Calibration curve of likelihood vs. regret for multi-stage SFT+MargE training.

