# OpenReview forum: "Generalists vs. Specialists: Evaluating LLMs on Highly-Constrained Biophysical Sequence Optimization Tasks"
_ICML.cc/2025/Conference — ICML 2025 poster_

### Official Review · Reviewer_xgMH · 2025-03-08

**Overall Recommendation:** 3

**Summary:**

This paper tackles the problem of biophysical sequence optimization - a task where even small deviations from stringent constraints (e.g., protein stability or solubility) can render a solution unusable. To bridge the gap between generalist LLM-based methods and specialist solvers, the authors introduce a synthetic test suite and a optimization framework that continuously optimize protein sequence using LLMs.

**Claims And Evidence:**

Yes, the proposed methods seem to be well-motivated and well supported.

**Essential References Not Discussed:**

N/A

**Ethics Expertise Needed:**

["Other expertise"]

**Experimental Designs Or Analyses:**

The overall experimental design is comprehensive and verify the effectiveness of  the proposed method

**Methods And Evaluation Criteria:**

Strengths:
- The paper presents a well-motivated and technically sound approach. The whole framework is self-contained, borrowing insights from discrete optimization, LLM fine-tuning, and preference learning.
- The benchmark (test suites) design is novel

Weakness/Question:
- Does the synthetic benchmarks generally applicable to real-world case?
- How is the computation cost of the LLMs?

**Other Comments Or Suggestions:**

N/A

**Other Strengths And Weaknesses:**

See Methods And Evaluation Criteria

**Questions For Authors:**

See Methods And Evaluation Criteria

**Relation To Broader Scientific Literature:**

The key contribution can be seen as a application of LLM on the tasks of biophysical sequence optimization, it would be interesting to researcher who works on this field

**Theoretical Claims:**

The paper does not contain any theoretical claims. The preference learning objectives are not new (in the appendix) so it doesn't really require further examination.

---

> ### Author Rebuttal · Authors · 2025-04-01
>
> Thank you for your positive assessment of our work. We appreciate your recognition that our approach is well-motivated and technically sound.
>
> ## On the applicability of synthetic benchmarks to real-world cases
>
> You raised an important question about whether our synthetic benchmarks apply to real-world cases. To address this directly, we conducted new experiments comparing optimizer performance on Ehrlich functions versus established lookup-based biological test functions (TFBind8, DhfR, TrpB -- see our response to reviewer b3Qx). We found strong rank correlations (0.61-0.89) between algorithm performance across these benchmarks, confirming that Ehrlich functions effectively capture the structure of real biological optimization problems.
>
> This validation supports our benchmark design principles, which were carefully crafted to capture key properties of real biophysical sequence optimization:
>
> 1. **Feasibility constraints**: The vast majority of random sequences fail to express or fold properly
> 2. **Epistasis**: Non-additive effects between sequence positions
> 3. **Position-dependent sensitivity**: The importance of specific residues at specific positions
> 4. **Motif constraints**: The need for functional motifs to appear with proper spacing
>
> By deliberately incorporating these properties, Ehrlich functions provide meaningful insights into algorithm performance on real-world biological optimization tasks while maintaining computational accessibility.
>
> ## On the computational cost of LLMs
>
> You also asked about the computational cost of using LLMs. We address this important practical consideration in Section 6.1:
>
> > "For relatively easy optimization problems, since the performance of various methods is similar, using a specialized model with 0.01% of the parameters of an LLM may be more practical."
>
> Our experiments revealed an interesting nuance: the optimal choice between generalist LLMs and specialist models depends on problem difficulty. For medium-difficulty problems, LLMs with appropriate training can significantly outperform specialized models, potentially justifying their higher computational cost. For very easy or very difficult problems, however, specialized models offer comparable performance with substantially lower computational requirements. This insight provides practical guidance for practitioners choosing between approaches based on their specific constraints and objectives.
>
> We have additionally conducted new LLOME-MargE experiments with a very small LLM (~226K params) trained from scratch (see our response to reviewer b3Qx). We found that even for a very small LLM with no pre-training, LLOME-MargE is significantly more sample efficient than LaMBO-2, despite having fewer model parameters than LaMBO-2. As such, the computational costs of LLOME need not always be a concern.
>
>
>
> Thank you again for your thoughtful review. We believe our work contributes valuable insights to both the machine learning and biological sequence design communities, and we appreciate your recognition of its potential impact.

---

### Official Review · Reviewer_2Up6 · 2025-03-13

**Overall Recommendation:** 3

**Summary:**

This paper investigates the use of large language models (LLMs) as black-box sequence optimizers for biophysical sequence design and optimization. The authors compare generalist LLM-based approaches with specialized optimization methods, such as LaMBO-2, to determine whether LLMs can efficiently optimize under strict biophysical constraints. The study introduces new benchmarks, novel training objectives, and a bilevel optimization framework to enhance LLM-based sequence optimization.

**Claims And Evidence:**

The majority of claims in this paper are supported by thorough experiments, well-defined benchmarks, and ablation studies. However, some claims would benefit from additional validation.

The only benchmark used is the Ehrlich functions, which are synthetic test functions, no real-world biological datasets (e.g., protein sequences, DNA regulatory elements) are tested.

**Essential References Not Discussed:**

No significant reference are missing.

**Experimental Designs Or Analyses:**

No issues.

**Methods And Evaluation Criteria:**

The proposed methods and evaluation criteria are generally well-designed for assessing LLMs in biophysical sequence optimization.

**Other Comments Or Suggestions:**

Please considering reference some prior work on the protein sequence optimization:

Chen, A., Stanton, S. D., Alberstein, R. G., Watkins, A. M., Bonneau, R., Gligorijević, V., ... & Frey, N. C. (2024). LLMs are highly-constrained biophysical sequence optimizers. arXiv preprint arXiv:2410.22296.

Gomez-Uribe, C. A., Gado, J., & Islamov, M. (2024). Designing diverse and high-performance proteins with a large language model in the loop. bioRxiv, 2024-10.

Subramanian, J., Sujit, S., Irtisam, N., Sain, U., Islam, R., Nowrouzezahrai, D., & Kahou, S. E. (2024). Reinforcement Learning for Sequence Design Leveraging Protein Language Models. arXiv preprint arXiv:2407.03154.

Wang, Y., He, J., Du, Y., Chen, X., Li, J. C., Liu, L. P., ... & Hassoun, S. (2025). Large Language Model is Secretly a Protein Sequence Optimizer. arXiv preprint arXiv:2501.09274.

**Other Strengths And Weaknesses:**

See above

**Questions For Authors:**

No

**Relation To Broader Scientific Literature:**

This paper contributes to LLM-based sequence optimization, preference learning for biophysical design, and comparisons between generalist (LLM-based) and specialist (model-based) solvers.

**Theoretical Claims:**

All correct.

---

> ### Author Rebuttal · Authors · 2025-04-01
>
> Thank you for your positive assessment and constructive feedback. We appreciate your recognition of our thorough experiments and well-defined benchmarks.
>
> ## On testing with real-world biological datasets
>
> You noted that our evaluation relies on synthetic Ehrlich functions rather than real-world biological datasets. While direct evaluation on real biological data would indeed be valuable, there are two key challenges:
>
> 1. **Training data contamination**: Using widely available biological datasets risks contamination with LLM training data, which would invalidate our assessment of LLMs as black-box optimizers.
>
> 2. **Accessibility**: Real biological datasets and simulators often require specialized software, significant computational or experimental resources, or domain expertise, creating barriers to reproduction and wider adoption.
>
> To address these concerns while maintaining biological relevance, we conducted new experiments comparing optimizer performance on Ehrlich functions versus established lookup-based biological test functions (TFBind8, DhfR, TrpB -- see our response to reviewer b3Qx). We found strong rank correlations (0.61-0.89) between algorithm performance across these benchmarks, confirming that Ehrlich functions effectively capture the structure of real biological optimization problems.
>
> This validation approach allows us to demonstrate biological relevance while avoiding contamination in LLM training data. It also preserves the computational accessibility that makes Ehrlich functions valuable for algorithm development.
>
> ## On additional references
>
> Thank you for suggesting additional relevant references. We will include them in the revised version to better situate our work within the growing literature on protein sequence optimization using LLMs.
>
> We believe our work provides several unique contributions to this field:
>
> 1. A systematically designed benchmark that balances realism, computational accessibility, and difficulty
> 2. A bilevel optimization framework that effectively leverages LLMs for constrained optimization
> 3. A novel preference learning objective that outperforms SFT, DPO, and REINFORCE when rewards are observed.
>
> These contributions advance both the theoretical understanding and practical application of LLMs for biological sequence optimization.
>
> Thank you again for your thoughtful review and suggestions for improvement.

---

### Official Review · Reviewer_mMvu · 2025-03-13

**Overall Recommendation:** 4

**Summary:**

The authors introduce Ehrlich functions, a novel synthetic function suite designed to simulate the properties of biological sequences and to facilitate benchmarking of generative algorithms for sequence optimization. They also propose a bilevel LLM-based solver, LLOME, which leverages a new preference loss called MargE. Experimental results, benchmarked against LAMBO-2 and GA, suggest that LLOME holds promise for biological sequence optimization tasks.

**Claims And Evidence:**

The authors’ claims regarding LLOME, Ehrlich functions, and the MargE preference loss are well substantiated by the experimental results, which clearly demonstrate the efficacy and potential of these methods for biological sequence optimization.

**Essential References Not Discussed:**

There are several other attempts in using protein language models in simulating the antibody maturation and experimentally validated the effectiveness with wet-lab experiments. The authors may discuss the links between the proposed method and the ones that biologiest are interested in. See https://doi.org/10.1038/s41587-023-01763-2 and DOI: 10.1126/science.adk8946

**Experimental Designs Or Analyses:**

One crucial consideration missing from the experimental design is the success rate. With unlimited time and resources, nearly any optimization method could eventually produce a sequence meeting the required criteria. However, in real-world settings—especially those involving costly wet-lab validation—it’s important to ensure a high success rate in the final (or a limited number of) round(s) of experiments.

**Methods And Evaluation Criteria:**

One key contribution of this work is the introduction of Ehrlich functions for evaluating generative algorithms in biological sequence optimization tasks. While these functions may oversimplify the complexity of real biological sequences, they provide a practical starting point, enabling rapid approximation and assessment of model performance.

**Other Comments Or Suggestions:**

See comments above.

**Other Strengths And Weaknesses:**

It appears that the authors rely on a single model for both scoring and generation. What are the advantages of using a unified approach compared to the more traditional setup in protein engineering, where one model serves as the ‘oracle’ (scoring function) and a separate LLM is responsible for sequence generation?

**Questions For Authors:**

Success rates: It would be highly informative to report the success rate of each optimization algorithm, as these metrics are critical for gauging practical utility—particularly when transitioning to costly wet-lab validation.

Ehrlich function applicability: Demonstrating whether Ehrlich functions can simulate real antibody maturation datasets would help validate their biological relevance. Such an evaluation could illustrate how well the functions capture key evolutionary or selection pressures inherent in the maturation process.

Protein language models in antibody maturation: A deeper discussion of how protein language models (PLMs) apply to antibody maturation would strengthen the manuscript’s impact.

One model vs 2 model: Explain the benefit in using one model for joint scoring and optimizing.

**Relation To Broader Scientific Literature:**

This work holds significant value for a broad range of scientific fields, including protein optimization, mRNA design, and antibody/CAR T-cell engineering. The design of Ehrlich functions is especially noteworthy for accelerating the development of advanced optimization algorithms. However, to strengthen its relevance for biologists, it would be helpful to validate these functions against existing antibody maturation datasets—demonstrating that, with proper parameter settings and initial seed sequences, they can closely simulate real biological processes. Additionally, expanding the discussion to address broader applications in other biological contexts could further underscore the method’s versatility.

**Theoretical Claims:**

Both the MargE and the construction of Ehrlich functions is sound and correct.

---

> ### Author Rebuttal · Authors · 2025-04-01
>
> Thank you for your positive assessment and thoughtful questions. We're pleased you recognize the value of our contributions and appreciate your suggestions for strengthening our work.
>
> ## On success rates and feasibility
>
> You raised an important point about success rates in real-world settings with costly wet-lab validation. We directly address this through our measurements of feasibility rates over time in Figures 4 and 10, which show how each method improves in generating valid sequences as optimization progresses.
>
> Figure 12 provides additional insight by analyzing the relationship between edit distance and feasibility, showing that LLOME-MargE achieves the best balance between exploration (making meaningful changes) and constraint satisfaction.
>
> Our simple regret plots (Figure 3) also demonstrate sample efficiency, which is critical when optimization is constrained by laboratory resources. LLOME-MargE consistently finds high-quality solutions with fewer function evaluations than other methods, particularly on medium-difficulty problems.
>
> ## On Ehrlich function applicability to real biology
>
> We share your interest in validating Ehrlich functions against real biological data. To address this, we conducted new experiments comparing optimizer performance on Ehrlich functions versus established biological test functions (TFBind8, DhfR, TrpB -- see the response to Reviewer b3Qx). We found strong rank correlations (0.61-0.89) between algorithm performance on these benchmarks, confirming that Ehrlich functions effectively capture the structure of real biological optimization problems.
>
> This validation approach allows us to demonstrate biological relevance while avoiding contamination in LLM training data, which could compromise benchmark integrity.
>
> ## On unified vs. separate models for scoring and generation
>
> You asked about the advantages of our unified approach compared to the traditional setup with separate scoring and generation models. The main benefits are:
>
> 1. **Improved ranking and guidance**: When a model is jointly trained on both tasks, its internal representations develop a better understanding of the relationship between sequence features and objective values. This improves the model's ability to generate and rank candidates effectively.
>
> 2. **Computational efficiency and search depth**: By unifying generation and evaluation in one model, we can perform deeper, more focused exploration with the same computational budget.
>
> 3. **Overcoming distributional limitations**: Traditional "generate-and-filter" approaches assume the desired outputs already exist in a mode of the generative model's training distribution. Scientific discovery inherently requires finding solutions in low-density regions or even outside the training distribution entirely. Our unified approach enables the model to progressively learn to generate such solutions.
>
> 4. **Simplified training and deployment**: Using a single model reduces engineering complexity and maintenance overhead in real-world applications.
>
> Thank you for your valuable suggestions. We agree that further validation on real antibody maturation datasets would strengthen our work's impact, and we're actively pursuing this direction for future research. We believe our current contributions provide a solid foundation for advancing the application of LLMs to biological sequence optimization problems.

---

> > ### Comment · Reviewer_mMvu · 2025-04-09
> >
> > Dear authors,
> >
> > Thank you for your comments and revisions. I still have concerns regarding the success rate -- it seems that the model use more than  20k evaluation steps to find a practical solution -- but in the work I mentioned, biologist can have nearly less than 24/96 "evaluation steps" to get a practical sequence. This could be an big issue. I hope more could be discussed here and if necessary, a small experiments for minimizing the gaps is important.
> >
> > Best,
> > mMvu

---

### Official Review · Reviewer_b3Qx · 2025-03-18

**Overall Recommendation:** 2

**Summary:**

This paper introduces a new synthetic test suite (Ehrlich functions) that captures the geometric structure of biophysical sequence optimization problems, proposes a framework LLOME (Language Model Optimization with Margin Expectation), a bilevel optimization routine for online black-box optimization, and uses a preference learning loss called MargE. To evaluate LLMs on biophysical sequence optimization, the paper conducts comparative evaluation of LLMs against specialized solvers like LaMBO-2.

**Claims And Evidence:**

Claims:
1. Off-the-shelf LLMs struggle to optimize Ehrlich functions with prompting alone (Yes)
2. LLOME with MargE can learn to solve some Ehrlich functions (Yes)
3. LLOME can outperform LaMBO-2 on moderately difficult Ehrlich variants (Yes)
4. LLMs show limited extrapolative capabilities without further training (Yes)

The evidence could be strengthened by including comparisons with more baselines.

**Essential References Not Discussed:**

More baseline works regarding LLMs and evolutionary algorithms can be considered. Here is a survey paper for reference: Evolutionary Computation in the Era of Large Language Model: Survey and Roadmap (https://arxiv.org/pdf/2401.10034)

There are other standard sequence optimization benchmarks like proteingym that may need to be considered too.

**Experimental Designs Or Analyses:**

Only two baselines are considered; it is not clear to me whether it is fair to compare the proposed LLOME on pretrained data with LAMBO-2 that is trained from random initialization.

**Methods And Evaluation Criteria:**

The proposed methods are reasonably sound for the problem as defined. Ehrlich functions provide a controlled environment for testing optimization capabilities with well-defined constraints. The evaluation criteria (regret metrics, feasibility, and diversity) are appropriate for measuring optimization performance.
However, the paper only compares against two baselines (genetic algorithm and LaMBO-2) and there is no evaluation on real biophysical optimization tasks to validate transferability.

**Other Comments Or Suggestions:**

See above

**Other Strengths And Weaknesses:**

Pros:
1. The paper identifies an important challenge in comparing general-purpose and specialized models
2. The analysis of preference learning calibration provides useful insights

Cons:
1. The significance of Ehrlich functions is not convincingly established; it's unclear why existing benchmarks weren't sufficient. It would be better to motivate why Ehrlich functions are specifically representative of biophysical sequence optimization rather than generic constrained optimization
2. The design of Ehrlich functions is not well motivated and not clearly explained in relation to biological sequence optimization tasks. It's unclear if the sequence nature of the problem is essential or just incidental to the optimization task
3. Only two baselines are compared and it seems the comparison is not fair with LAMBO-2

**Questions For Authors:**

1. How do the authors justify that performance on Ehrlich functions would translate to real biophysical sequence optimization tasks? Could the authors provide evidence of correlation between performance on Ehrlich functions and established benchmarks?
2. Why develop a new synthetic benchmark rather than using established ones like those in ProteinGym? What specific limitations of existing benchmarks necessitated this approach?
3. Given that LLMs require significantly more computational resources, how would the authors characterize the trade-offs between performance and efficiency when choosing between generalist and specialist approaches? At what point would the performance improvements justify the increased computational costs?

**Relation To Broader Scientific Literature:**

The paper builds on directed evolution and genetic algorithms for black-box optimization, adding LLM-based approaches to this domain.

**Theoretical Claims:**

The derivation of MargE in Appendix A.3 appears mathematically sound, though relatively straightforward compared to prior work in preference learning. The proof of Lemma A.1 establishing properties of Bradley-Terry models with reward functions helps justify the design choices in the reward function but is not especially novel.

---

> ### Author Rebuttal · Authors · 2025-04-01
>
> Thank you for your thoughtful assessment of our work. We appreciate your recognition of our technical contributions and would like to address your concerns.
>
> ## On the choice of Ehrlich functions over existing benchmarks
>
> To validate the real-world applicability of Ehrlich functions, we conducted new experiments comparing optimizer performance on Ehrlich functions vs. 3 established lookup-based test functions:
> - [TFBind8](https://www.science.org/doi/10.1126/science.aad2257): DNA transcription factor binding optimization (8-base sequence)
> - [TrpB](https://www.science.org/doi/10.1126/science.adh3860): Tryptophan synthase β-subunit protein optimization (4-amino acid sequence)
> - [DhfR](https://www.pnas.org/doi/10.1073/pnas.2400439121): Dihydrofolate reductase DNA binding optimization (9-base sequence)
>
> For each benchmark, we evaluated the median cumulative regret (estimated from 8 trials) of 64 variants of our GA with different hyperparameter settings. We then computed rank correlations between algorithm performance on these biological benchmarks vs. comparable Ehrlich functions. The strong rank correlations confirm that Ehrlich functions effectively capture the structure of real biological optimization problems:
> | Spearman Corr. with → | DhfR | TFBind8 | TrpB |
> |--------------------|------|---------|------|
> | Ehr(4, 4)-2-2-2    | 0.75 | 0.75    | 0.61 |
> | Ehr(20, 8)-2-2-2   | 0.86 | 0.89    | 0.73 |
>
> ## On the design of Ehrlich functions
> We developed Ehrlich functions after carefully analyzing the limitations of existing benchmarks. As we explain in Section A.2.2, current benchmarks fall into several categories, each with significant drawbacks. **Database lookup** benchmarks are costly to construct and unnecessarily restrictive of the search space. **Empirical function approximations** often have spurious optima that are easy to find but not reflective of real solutions for the biological problem. **Physics-based simulations** are slow to evaluate, difficult to install/run correctly, and admit trivial solutions that score well but are not desirable.
>
> Instead, we designed the Ehrlich suite to have the following criteria: (1) low compute cost, (2) well-characterized solutions, (3) non-trivial difficulty, (4) similarity to real-life applications, and (5) not already seen in training data. To the best of our knowledge, this is currently the only biosequence optimization benchmark to possess all 5 attributes.
>
> ## On the relationship between Ehrlich functions and biological sequence optimization
> Ehrlich functions capture four key properties of real biophysical sequence optimization:
> 1. **Feasibility constraints**: The vast majority of random sequences are non-viable/non-expressible
> 2. **Epistasis**: Non-additive effects between sequence positions
> 3. **Position-dependent sensitivity**: The importance of specific residues at specific positions
> 4. **Motif constraints**: The need for functional motifs to appear with proper spacing
> As we illustrate in Fig. 6, these properties are directly related to antibody-antigen binding.
>
> ## On the choice of comparisons
> We have added a comparison against LLOME-MargE with a smaller LLM (~226K params), trained from scratch. Although pre-training should not offer any additional advantages due to the lack of overlap between Ehrlich functions and the pre-training data, we choose this setting to be similar in model size and training to LaMBO-2. We evaluate on the f2 function (i.e. Ehr(32, 32)-4-4-4). Despite being ~2/3 the size of LaMBO-2, this model achieves the same min. regret using several thousand fewer test function evaluations (please compare to LaMBO-2 results in Fig. 3b):
> | # Test Function Evals |   10000 |   12000 |   14000 |   16000 |   18000 |   20000 |   22000 |   24000 |   26000 |   28000 |
> |:-----------|--------:|--------:|--------:|--------:|--------:|--------:|--------:|--------:|--------:|--------:|
> | Min. Regret |   0.906 |   0.859 |   0.789 |   0.719 |   0.625 |     0.5 |   0.438 |   0.438 |   0.438 |   0.438 |
>
> This illustrates the strength of LLOME-MargE, even in small models without any pre-training.
>
> ## On trade-offs
> You raise an important question about the trade-offs between LLMs and specialized models. We explicitly address this in Section 6.1:
> > "For relatively easy optimization problems, since the performance of various methods is similar, using a specialized model with 0.01% of the parameters of an LLM may be more practical."
>
> Our results show that the choice between generalist and specialist models depends - for medium-difficulty problems, the performance improvements of LLMs may justify their cost. For very easy or difficult problems, specialized models offer comparable performance with much less compute.
>
> ## On essential references
> We have an extended Related Work section in A.1, which discusses many LLM + evolution works. Also, ProteinGym is not a sequence generation benchmark.

---

### Decision · Program_Chairs · 2025-05-01

**Decision:**

Accept (poster)

**Comment:**

The authors introduce a synthetic benchmark suite and LLM-based optimization algorithm that does well on this suite. The reviewers overall appreciate the contributions of the paper, and the synthetic benchmark suite in particular. I do think that there are some lingering weaknesses -- namely that it would be great to validate the synthetic benchmark suite against empirical data to show that performance on the synthetic benchmark suite correlates with some real world task of interest. To this end I do appreciate the authors' inclusion of rank correlations against TFBind8, DhfR, TrpB (and am ultimately recommending acceptance), although I do think as much validation here would be great.